# The Complexity of Learning Sparse Superposed Features with Feedback

**Akash Kumar** [1]

## Abstract

The success of deep networks is crucially attributed to their ability to capture latent features within a representation space. In this work, we investigate whether the underlying learned features of a model can be efficiently retrieved through feedback from an agent, such as a large language model (LLM), in the form of relative *triplet comparisons*. These features may represent various constructs, including dictionaries in LLMs or a covariance matrix of Mahalanobis distances. We analyze the feedback complexity associated with learning a feature matrix in sparse settings. Our results establish tight bounds when the agent is permitted to construct activations and demonstrate strong upper bounds in sparse scenarios when the agent's feedback is limited to distributional information. We validate our theoretical findings through experiments[1] on two distinct applications: feature recovery from Recursive Feature Machines and dictionary extraction from sparse autoencoders trained on Large Language Models.

## 1. Introduction

In recent years, neural network-based models have achieved state-of-the-art performance across a wide array of tasks. These models effectively capture relevant features or concepts from samples, tailored to the specific prediction tasks they address (Yang and Hu, 2021b; Bordelon and Pehlevan, 2022a; Ba et al., 2022b). A fundamental challenge lies in understanding how these models learn such features and determining whether these features can be interpreted or even retrieved directly (Radhakrishnan et al., 2024). Recent advancements in *mechanistic interpretability* have opened

multiple avenues for elucidating how transformer-based models, including Large Language Models (LLMs), acquire and represent features (Bricken et al., 2023; Doshi-Velez and Kim, 2017). These advances include uncovering neural circuits that encode specific concepts (Marks et al., 2024b; Olah et al., 2020), understanding feature composition across attention layers (Yang and Hu, 2021b), and revealing how models develop structured representations (Elhage et al., 2022). One line of research posits that features are encoded linearly within the latent representation space through sparse activations, a concept known as the linear representation hypothesis (LRH) (Mikolov et al., 2013; Arora et al., 2016). However, this hypothesis faces challenges in explaining how neural networks function, as models often need to represent more distinct features than their layer dimensions would theoretically allow under purely linear encoding. This phenomenon has been studied extensively in the context of large language models through the lens of superposition (Elhage et al., 2022), where multiple features share the same dimensional space in structured ways.

Recent efforts have addressed this challenge through sparse coding or dictionary learning, proposing that any layer $\ell$ of the model learns features linearly:

$$\boldsymbol{x} \approx \mathsf{D}_\ell \cdot \alpha_\ell(\boldsymbol{x}) + \epsilon_\ell(\boldsymbol{x}),$$

where $\boldsymbol{x} \in \mathbb{R}^d$, $\mathsf{D}_\ell \in \mathbb{R}^{d \times p}$ is a dictionary[2] matrix, $\alpha_\ell(\boldsymbol{x}) \in \mathbb{R}^p$ is a sparse representation vector, and $\epsilon_\ell(\boldsymbol{x}) \in \mathbb{R}^p$ represents error terms. This approach enables retrieval of interpretable features through sparse autoencoders (Bricken et al., 2023; Marks et al., 2024b), allowing for targeted monitoring and modification of network behavior. The linear feature decomposition not only advances model interpretation but also suggests the potential for developing compact, interpretable models that maintain performance by leveraging universal features from larger architectures.

In this work, we explore how complex features encoded as a dictionary can be distilled through feedback from either advanced language models (e.g., ChatGPT, Claude 3.0 Sonnet) or human agents. Let's define a dictionary $\mathsf{D} \in \mathbb{R}^{d \times p}$ where each column represents an atomic feature vector. These atomic features, denoted as $u_1, u_2, \ldots, u_p \subset \mathbb{R}^d$, could correspond to semantic concepts like "tree", "house", or "lawn" that are relevant to the task's sample

---

[*]Equal contribution [1]Department of Computer Science & Engineering, University of California, San Diego, USA. Correspondence to: Akash Kumar <akk002@ucsd.edu>.

*Proceedings of the $42^{nd}$ International Conference on Machine Learning*, Vancouver, Canada. PMLR 267, 2025. Copyright 2025 by the author(s).

[1](https://github.com/akashkumar-d/
learnsparsefeatureswithfeedback.git)

---

[2]could be both overcomplete and undercomplete.

space. The core mechanism involves an agent (either AI or human) determining whether different sparse combinations of these atomic features are similar or dissimilar. Specifically, given sparse activation vectors $\alpha, \alpha' \in \mathbb{R}^p$, the agent evaluates whether linear combinations such as $\alpha_1 v(\text{"tree"}) + \alpha_2 v(\text{"car"}) + ... + \alpha_d v(\text{"house"})$ are equivalent to other combinations using different activation vectors. Precisely, we formalize these feedback relationships using relative triplet comparisons $(\alpha, \beta, \zeta) \in \mathcal{V}$, where $\mathcal{V} \subseteq \mathbb{R}^p$ is the activation or representation space. These comparisons express that a linear combination of features using coefficients $\alpha$ is more similar to a combination using coefficients $\beta$ than to one using coefficients $\zeta$.

The objective is to determine the extent to which an oblivious learner—one who learns solely by satisfying the constraints of the feedback and randomly selecting valid features—can identify the feature vectors of $\mathsf{D}$ up to *normal transformation*. The fundamental protocol is as follows:

- The agent either constructs or selects (from a sampled pool) sparse triplets of activations $(\alpha, \beta, \zeta) \in \mathbb{R}^{3p}$ and designs relative feedback of similarity $\ell \in \{+1, 0, -1\}$ satisfying $\mathrm{sgn}\left(\|\mathsf{D}(\alpha - \beta)\| - \|\mathsf{D}(\alpha - \zeta)\|\right) = \ell$, and provides them to the learner.
- The learner solves for
$$\left\{ \mathrm{sgn}\left( \|\hat{\mathsf{D}}(\alpha - \beta)\| - \|\hat{\mathsf{D}}(\alpha - \zeta)\| \right) = \ell \right\}$$
and outputs a solution $\hat{\mathsf{D}}^\top \hat{\mathsf{D}}$.

Semantically, these relative distances provide the relative information on how ground truth samples, e.g. images, text among others, relate to each other. We term the normal transformation $\mathsf{DD}^\top$ for a given dictionary $\mathsf{D}$ as feature matrices $\mathbf{\Phi} \in \mathbb{R}^{p \times p}$, which is exactly a covariance matrix. Alternatively, for the representation space $\mathcal{V} \subseteq \mathbb{R}^p$, this transformation defines a Mahalanobis distance function $d : \mathcal{V} \times \mathcal{V} \to \mathbb{R}$, characterized by the square symmetric linear transformation $\mathbf{\Phi} \succeq 0$ such that for any pair of activations $(x, y) \in \mathcal{V}^2$, their distance is given by:
$$d(x, y) := (x - y)^\top \mathbf{\Phi} (x - y)$$

When $\mathbf{\Phi}$ embeds samples into $\mathbb{R}^r$, it admits a decomposition $\mathbf{\Phi} = \mathsf{L}^\top \mathsf{L}$ for $\mathsf{L} \in \mathbb{R}^{r \times p}$, where $\mathsf{L}$ serves as a dictionary for this distance function—a formulation well-studied in metric learning literature (Kulis, 2013). In this work, we study the minimal number of interactions, termed as feedback complexity of learning feature matrices—normal transformations to a dictionary—of the form $\mathbf{\Phi}^* \in \mathsf{Sym}_+(\mathbb{R}^{p \times p})$. We consider two types of feedback: general activations and sparse activations, examining both constructive and distributional settings. Our primary contributions are:

I. We investigate feedback complexity in the constructive setting, where agents select activations from $\mathbb{R}^p$, establishing strong bounds for both general and sparse scenarios. (see Section 4)

II. We analyze the distributional setting with sampled activations, developing results for both general and sparse representations. For sparse sampling, we extend the definition of a Lebesgue measure to accommodate sparsity constraints. (see Section 5)

III. We validate our theoretical bounds through experiments with feature matrices from Recursive Feature Machines and dictionaries trained for sparse autoencoders in Large Language Models, including Pythia-70M (Biderman et al., 2023) and Board Game models (Karvonen et al., 2024). (see Section 6)

Table 1 summarizes our feedback complexity bounds.

## 2. Related Work

**Dictionary learning** Recent work has explored dictionary learning to disentangle the semanticity (mono- or polysemy) of neural network activations (Faruqui et al., 2015; Arora et al., 2018; Zhang et al., 2019; Yun et al., 2021). Dictionary learning (Mallat and Zhang, 1993; Olshausen and Field, 1997) (aka sparse coding) provides a systematic approach to decompose task-specific samples into sparse signals. The sample complexity of dictionary learning (or sparse coding) has been extensively studied as an optimization problem, typically involving non-convex objectives such as $\ell_1$ regularization (see (Gribonval et al., 2015)). While traditional methods work directly with ground-truth samples, our approach differs fundamentally as the learner only receives feedback on sparse signals or activations. Prior work in noiseless settings has established probabilistic exact recovery up to linear transformations (permutations and sign changes) under mutual incoherence conditions (Gribonval and Schnass, 2010; Agarwal et al., 2014). Our work extends these results by proving exact recovery (both deterministic and probabilistic) up to normal transformation, which generalizes to rotational and sign changes under strong incoherence properties (see Lemma 1). In the sampling regime, we analyze $k$-sparse signals, building upon the noisy setting framework developed in Arora et al. (2013); Gribonval et al. (2015).

**Feature learning in neural networks and Linear representation hypothesis** Neural networks demonstrate a remarkable ability to discover and exploit task-specific features from data (Yang and Hu, 2021b; Bordelon and Pehlevan, 2022b; Shi et al., 2022). Recent theoretical advances have significantly enhanced our understanding of feature evolution and emergence during training (Abbe et al., 2022; Ba et al., 2022a; Damian et al., 2022; Yang and Hu, 2021a; Zhu et al., 2022). Particularly noteworthy is the finding that the outer product of model weights correlates with the gradient outer product of the classifier averaged over layer preactivations (Radhakrishnan et al., 2024), which directly relates to the covariance matrices central to our investigation. Building upon these insights, Elhage et al.

| Feedback type | Standard Constructive | Sparse Constructive | Standard Sampling | Sparse Sampling |
|---|---|---|---|---|
| **Feedback Complexity** | $\Theta\left(\frac{r(r+1)}{2} + p - r + 1\right)$ | $O\left(\frac{p(p+1)}{2}\right)$ | $^*\Theta\left(\frac{p(p+1)}{2}\right)$ | $^*c\, p^2 \left(\frac{2}{p_{\mathsf{s}}^2} \log \frac{2}{\delta}\right)^{\frac{1}{p^2}}$ |

| Prior Works | SAE (Sharkey et al., 2025) | CRAFT (Fel et al., 2023) | Probing (Marks and Tegmark, 2024) | | |
|---|---|---|---|---|---|
| | | | LR | CCS (Burns et al., 2023) | LDA |
| **Learning Complexity** | $Tnpd$ | $npk$ | $Tnp$ | $Tnp$ | $\mathcal{O}(np^2 + p^3)$ |

Table 1: Comparison of feedback complexity in this work against prior feature retrieval (learning) methods. $T$: number of iterations, $n$: number of samples, $p$: activation dimension, $d$: input space dimension, $k$: number of latent components, $r$: the rank of the feature matrix, $c > 0$ is a constant, and $p_{\mathsf{s}}$ depends on activation distribution and sparsity $s$. We use $^*$ to denote "almost surely" and $^*$ to denote "with high probability" guarantees.

(2022) proposed that features in large language models follow a linear encoding principle, suggesting that the complex feature representations learned during training can be decomposed into interpretable linear components. This interpretability, in turn, could facilitate the development of simplified algorithms for complex tasks (Fawzi et al., 2022; Romera-Paredes et al., 2024). Recent research has focused on extracting these interpretable features in the form of dictionary learning by training sparse autoencoder for various language models including Board Games Models (Marks et al., 2024b; Bricken et al., 2023). Our work extends this line of inquiry by investigating whether such interpretable dictionaries can be effectively transferred to a weak learner using minimal comparative feedback.

**Triplet learning a covariance matrix**   Learning a feature matrix (for a dictionary) up to normal transformation can be viewed through two established frameworks: covariance estimation (Chen et al., 2013; Li and Voroninski, 2013) and learning Mahalanobis distances (Kulis, 2013). While these frameworks traditionally rely on exact or noisy measurements, our work introduces a distinct mechanism based solely on relative feedback, aligning more closely with the semantic structure of Mahalanobis distances. The study of such distances has been central to metric learning research (Bellet et al., 2015; Kulis, 2013), encompassing both supervised approaches (Weinberger and Saul, 2009; Xing et al., 2002) and unsupervised methods such as LDA (Fisher, 1936) and PCA (Jolliffe, 1986). Schultz and Joachims (2003) and Kleindessner and von Luxburg (2016) have extended this framework to incorporate relative comparisons on distances. Particularly relevant to our work are studies by Schultz and Joachims (2003) and Mason et al. (2017) that employ triplet comparisons, though these typically assume i.i.d. triplets with potentially noisy measurements. Our approach differs by incorporating an active learning element: while signals are drawn i.i.d, an agent selectively provides feedback on informative instances. This constructive triplet framework for covariance estimation represents a novel direction, drawing inspiration from machine teaching, where a teaching agent provides carefully chosen examples to facilitate learning (Zhu et al., 2018; Kumar et al., 2021).

## 3. Problem Setup

We denote by $\mathcal{V} \subseteq \mathbb{R}^p$ the space of activations or representations and by $\mathcal{X} \subseteq \mathbb{R}^d$ the space of samples. For the space of feature matrices (for a dictionary or Mahalanobis distances), denoted as $\mathcal{M}_{\mathsf{F}}$, we consider the family of symmetric positive semi-definite matrices in $\mathbb{R}^{p \times p}$, i.e. $\mathcal{M}_{\mathsf{F}} = \left\{ \Phi \in \mathsf{Sym}_+(\mathbb{R}^{p \times p}) \right\}$. We denote a feedback set as $\mathcal{F}$ which consists of triplets $(x, y, z) \in \mathcal{V}^3$ with corresponding signs $\ell \in \{+1, 0, -1\}$.

We use the standard notations in linear algebra over a space of matrices provided in Appendix B.

**Triplet feedback**   An agent provides feedback on activations in $\mathcal{V}$ through relative triplet comparisons $(x, y, z) \in \mathcal{V}$. Each comparison evaluates linear combinations of feature vectors:

$$\sum_{i=1}^{p} x_i u_i(\text{"feature } i\text{"}) \text{ is more similar to}$$

$$\sum_{i=1}^{p} y_i u_i(\text{"feature } i\text{"}) \text{ than to } \sum_{i=1}^{p} z_i u_i(\text{"feature } i\text{"})$$

We study both sparse and non-sparse activation feedbacks, where sparsity is defined as:

**Definition 1** (*s*-sparse activations)**.** *An activation $\alpha \in \mathbb{R}^p$ is $s$-sparse if at most $s$ many indices of $\alpha$ are non-zero.*

Since triplet comparisons are invariant to positive scaling of feature matrices, we define:

**Definition 2** (Feature equivalence)**.** *For a feature family $\mathcal{M}_{\mathsf{F}}$, feature matrices $\Phi'$ and $\Phi^*$ are equivalent if there exists $\lambda > 0$ such that $\Phi' = \lambda \cdot \Phi^*$.*

We study a learning framework where the learner merely satisfies the constraints provided by the agent's feedback:

**Definition 3** (Oblivious learner)**.** *A learner is oblivious if it randomly selects a feature matrix from the set of valid solutions to a given feedback set $\mathcal{F}$, i.e., arbitrarily chooses $\Phi \in \mathcal{M}_{\mathsf{F}}(\mathcal{F})$, where $\mathcal{M}_{\mathsf{F}}(\mathcal{F})$ represents the set of feature matrices satisfying the constraints in $\mathcal{F}$.*

This framework aligns with version space learning, where $\mathsf{VS}(\mathcal{F}, \mathcal{M}_{\mathsf{F}})$ denotes the set of feature matrices in $\mathcal{M}_{\mathsf{F}}(\mathcal{F})$ compatible with feedback set $\mathcal{F}$.



Figure 1: **Features via Recursive Feature Machines.** We perform monomial regression on $z \sim \mathcal{N}(0, 0.5I_{10})$ with target $f^*(z) = z_0 \, z_1 \, \mathbf{1}(z_5 > 0)$. An RFM kernel machine $\hat{f}_{\mathbf{\Phi}}(z) = \sum_{y_i \in \mathcal{D}_{\text{train}}} a_i K_{\mathbf{\Phi}}(y_i, z)$ is trained for 5 iterations on 4000 samples to produce the ground-truth feature matrix $\mathbf{\Phi}^*$ of rank 4 (Radhakrishnan et al., 2024). We then query an agent for feedback via: eigendecomposition (Theorem 1), sparse constructive (Theorem 2), random Gaussian sampling (Theorem 3), and sparse sampling with $\mu = 0.9$ (Theorem 4). Eigendecomposition, sparse constructive, and random sampling achieve the ground-truth MSE with only 55 feedbacks, whereas high-sparsity sampling yields inferior features and larger MSE.

Prior work on dictionary learning has established recovery up to linear transformation under weak mutual incoherence (Gribonval and Schnass, 2010). In our setting, with the agent's feature feedback corresponding to $\mathsf{D}$ (or $\mathsf{L}$) $\in \mathbb{R}^{d \times p}$, the learner recovers $\mathsf{L}$ up to normal transformation. Moreover, when $\mathsf{L}$ has orthogonal rows (strong incoherence), we can recover $\mathsf{L}$ up to rotation and sign changes as stated below, with proof deferred to Appendix D.

**Lemma 1** (Recovering orthogonal representations). *Assume* $\mathbf{\Phi} \in \mathsf{Sym}_+(\mathbb{R}^{p \times p})$ . *Define the set of orthogonal Cholesky decompositions of $\mathbf{\Phi}$ as*

$$\mathcal{W}_{\mathsf{CD}} = \left\{ \boldsymbol{U} \in \mathbb{R}^{p \times r} \,\middle|\, \mathbf{\Phi} = \boldsymbol{U}\boldsymbol{U}^\top \,\&\, \boldsymbol{U}^\top \boldsymbol{U} = diag(\lambda_1, \ldots, \lambda_r) \right\},$$

*where $r = rank(\mathbf{\Phi})$ and $\lambda_1, \lambda_2, \ldots, \lambda_r$ are the eigenvalues of $\mathbf{\Phi}$ in descending order. Then, for any two matrices $\boldsymbol{U}, \boldsymbol{U}' \in \mathcal{W}_{\mathsf{CD}}$, there exists an orthogonal matrix $\boldsymbol{R} \in \mathbb{R}^{r \times r}$ such that $\boldsymbol{U}' = \boldsymbol{U}\boldsymbol{R}$, where $\boldsymbol{R}$ is block diagonal with orthogonal blocks corresponding to any repeated diagonal entries $\lambda_i$ in $\boldsymbol{U}^\top \boldsymbol{U}$. Additionally, each column of $\boldsymbol{U}'$ can differ from the corresponding column of $\boldsymbol{U}$ by a sign change.*

We note that the recovery of $\mathsf{L}$ is pertaining to the assumption that all the rows are orthogonal, and thus rank of $\mathsf{L}$ is $r = d$. In cases where $r < d$, one needs additional information in the form of ground sample $x = \mathsf{L}\alpha$ for some activation $\alpha$ to recover $\mathsf{L}$ up to a linear transformation. Finally, the interaction protocol is shown in Algorithm 1.

## 4. Sparse Feature Learning with Constructive Feedback

Here, we study the feedback complexity in the setting where agent is allowed to pick/construct any activation from $\mathbb{R}^p$.

**Reduction to Pairwise Comparisons** The general triplet feedbacks with potentially inequality constraints in Algorithm 1 can be simplified to pairwise comparisons with equality constraints with a simple manipulation as follows.

---

**Algorithm 1** Model of Feature learning with feedback

**Given**: Representation space $\mathcal{V} \subseteq \mathbb{R}^p$, Feature family $\mathcal{M}_\mathsf{F}$

*In batch setting:*

1. Teacher picks triplets $\mathcal{F}(\mathcal{V}, \mathbf{\Phi}^*) =$
$\left\{ (x, y, z) \in \mathcal{V}^3 \,\middle|\, (x-y)^\top \mathbf{\Phi}^*(x-y) \geq (x-z)^\top \mathbf{\Phi}^*(x-z) \right\}$

2. Learner receives $\mathcal{F}$, and obliviously picks a feature matrix $\mathbf{\Phi} \in \mathcal{M}_\mathsf{F}$ that satisfy the set of constraints in $\mathcal{F}(\mathcal{V}, \mathbf{\Phi}^*)$

3. Learner outputs $\mathbf{\Phi}$.

---

**Lemma 2.** *Let $\mathbf{\Phi}^* \in \mathcal{M}_\mathsf{F}$ be a target feature matrix in representation space $\mathbb{R}^p$ used for oblivious learning. Given a feedback set*

$$\mathcal{F} = \{(x, y, z) \in \mathbb{R}^{3p} \,|\, (x-y)^\top \mathbf{\Phi}^*(x-y) \geq (x-z)^\top \mathbf{\Phi}^*(x-z)\},$$

*such that any $\mathbf{\Phi}' \in \mathsf{VS}(\mathcal{F}, \mathcal{M}_\mathsf{F})$ is feature equivalent to $\mathbf{\Phi}^*$, there exists a pairwise feedback set*

$$\mathcal{F}' = \left\{ (y', z') \in \mathbb{R}^{2p} \,\middle|\, y'^\top \mathbf{\Phi}^* y' = z'^\top \mathbf{\Phi}^* z' \right\}$$

*such that $\mathbf{\Phi}' \in \mathsf{VS}(\mathcal{F}', \mathcal{M}_\mathsf{F})$.*

*Proof.* WLOG, assume $x \neq z$ for all $(x, y, z) \in \mathcal{F}$. For any triplet $(x, y, z) \in \mathcal{F}$: **Case** (i): If $(x-y)^\top \mathbf{\Phi}^*(x-y) = (x-z)^\top \mathbf{\Phi}^*(x-z)$, then $(x-y, x-z)$ satisfies the equality. **Case** (ii): If $(x-y)^\top \mathbf{\Phi}^*(x-y) > (x-z)^\top \mathbf{\Phi}^*(x-z)$, then for some $\lambda > 0$:

$$(x-y)^\top \mathbf{\Phi}^*(x-y) = (1+\lambda)(x-z)^\top \mathbf{\Phi}^*(x-z)$$

implying $(x-y, \sqrt{1+\lambda}(x-z))$ satisfies the equality.

Thus, each triplet in $\mathcal{F}$ maps to a pair in $\mathcal{F}'$, preserving feature equivalence under positive scaling. $\square$

This implies that if triplet comparisons are used in Algorithm 1, equivalent pairwise comparisons exist satisfying:

$$\mathbf{\Phi}' = \lambda \cdot \mathbf{\Phi}^*, \quad \lambda > 0, \tag{1a}$$

$$\mathbf{\Phi}' \in \left\{ \mathbf{\Phi} \in \mathcal{M}_{\mathsf{F}} \,\middle|\, \forall (y, z) \in \mathcal{F}', \, y^\top \mathbf{\Phi} y = z^\top \mathbf{\Phi} z \right\}. \tag{1b}$$

Now, we show a reformulation of the oblivious learning problem for a feature matrix using pairwise comparisons that provide a unique geometric interpretation. Consider a pair $(y, z)$ and a matrix $\mathbf{\Phi}$. An equality constraint implies

$$y^\top \mathbf{\Phi} y = z^\top \mathbf{\Phi} z \iff \langle \mathbf{\Phi}, yy^\top - zz^\top \rangle = 0$$

where $\langle \cdot, \cdot \rangle$ denotes the Frobenius inner product. Now, given a set of pairwise feedbacks

$$\mathcal{F}(\mathbb{R}^p, \mathcal{M}_{\mathsf{F}}, \mathbf{\Phi}^*) = \{(y_i, z_i)\}_{i=1}^k$$

corresponding to the target feature matrix $\mathbf{\Phi}^*$, the learning problem defined by Eq. (1b) can be formulated as:

$$\forall (y, z) \in \mathcal{F}(\mathbb{R}^p, \mathcal{M}_{\mathsf{F}}, \mathbf{\Phi}^*), \quad \langle \mathbf{\Phi}, yy^\top - zz^\top \rangle = 0. \tag{2}$$

Geometrically, the condition in Eq. (2) implies that any solution $\mathbf{\Phi}$ should annihilate the subspace of the orthogonal complement that is spanned by the matrices $\{yy^\top - zz^\top\}_{(y,z)\in\mathcal{F}}$. Formally, this complement is defined as:

$$\mathcal{O}_{\mathbf{\Phi}^*} := \left\{ S \in \mathsf{Sym}(\mathbb{R}^{p\times p}) \,\middle|\, \langle \mathbf{\Phi}^*, S \rangle = 0 \right\}.$$

### 4.1. Constructive feedbacks: Worst-case lower bound

To learn a symmetric PSD matrix, learner needs at most $p(p+1)/2$ constraints for linear programming corresponding to the number of degrees of freedom. So, the first question is are there pathological cases of feature matrices in $\mathcal{M}_{\mathsf{F}}$ which would require at least $p(p+1)/2$ many triplet feedbacks in Algorithm 1. This indeed is the case, if a target matrix $\mathbf{\Phi}^* \in \mathsf{Sym}_+(\mathbb{R}^{p\times p})$ is full rank.

In the following proposition proven in Appendix E, we show a strong lower bound on the worst-case $\mathbf{\Phi}^*$ that turns out to be of order $\Omega(p^2)$.

**Proposition 1.** *In the constructive setting, the worst-case feedback complexity of the class $\mathcal{M}_{\mathsf{F}}$ with general activations is at the least $(p(p+1)/2 - 1)$.*

*Proof Outline.* As discussed in Eq. (1) and Eq. (2), for a full-rank feature matrix $\mathbf{\Phi}^* \in \mathcal{M}_{\mathsf{F}}$, the span of any feedback set $\mathcal{F}$, i.e., $\mathrm{span}\langle\{xx^\top - yy^\top\}_{(x,y)\in\mathcal{F}}\rangle$, must lie within the orthogonal complement $\mathcal{O}_{\mathbf{\Phi}^*}$ of $\mathbf{\Phi}^*$ in the space of symmetric matrices $\mathsf{Sym}(\mathbb{R}^{p\times p})$. Conversely, if $\mathbf{\Phi}^*$ has full rank, then $\mathcal{O}_{\mathbf{\Phi}^*}$ is contained within this span. This necessary condition requires the feedback set to have a size of at least $\frac{p(p+1)}{2} - 1$, given that $\dim(\mathsf{Sym}(\mathbb{R}^{p\times p})) = \frac{p(p+1)}{2}$. $\square$

Since the worst-case bound is pessimistic for oblivious learning of Eq. (1) a general question is how feedback complexity varies over the feature model $\mathcal{M}_{\mathsf{F}}$. Now, we study the feedback complexity for feature model based on the rank of the matrix, showing that the bounds can be drastically reduced.

### 4.2. Feature learning of low-rank matrices

As stated in Proposition 1, the learner requires at least $\frac{p(p+1)}{2} - 1$ feedback pairs to annihilate the orthogonal complement $\mathcal{O}_{\mathbf{\Phi}^*}$. However, this requirement decreases with a lower rank of $\mathbf{\Phi}^*$. We illustrate this in Fig. 1 for a feature matrix $\mathbf{\Phi} \in \mathbb{R}^{10\times10}$ of rank 4 trained via Recursive Feature Machines (Radhakrishnan et al., 2024).

Consider an activation $\alpha \in \mathbb{R}^p$ in the nullspace of $\mathbf{\Phi}^*$. Since $\mathbf{\Phi}^*\alpha = 0$, it follows that $\alpha^\top\mathbf{\Phi}^*\alpha = 0$. Moreover, for another activation $\beta \notin \mathrm{span}\langle\alpha\rangle$ in the nullspace, any linear combination $a\alpha + b\beta$ satisfies

$$(a\alpha + b\beta)^\top \mathbf{\Phi}^* (a\alpha + b\beta) = 0.$$

This suggests a strategy for designing effective feedback based on the kernel $\mathrm{Ker}(\mathbf{\Phi}^*)$ and the null space $\mathrm{null}(\mathbf{\Phi}^*)$ of $\mathbf{\Phi}^*$ (see Appendix B for table of notations). This intuition is formalized by the eigendecomposition of the feature matrix:

$$\mathbf{\Phi}^* = \sum_{i=1}^r \lambda_i u_i u_i^\top, \tag{3}$$

where $\{\lambda_i\}$ are the eigenvalues and $\{u_i\}$ are the orthonormal eigenvectors. Since $\mathbf{\Phi}^* \succeq 0$ this decomposition is *unique* with non-negative eigenvalues.

To teach $\mathbf{\Phi}^*$, the agent can employ a dual approach: teaching the kernel associated with the eigenvectors in this decomposition and the null space separately. Specifically, the agent can provide feedbacks corresponding to the eigenvectors of $\mathbf{\Phi}^*$'s kernel and extend the basis $\{u_i\}$ for the null space. We first present the following useful result (see proof in Appendix F).

**Lemma 3.** *Let $\{v_i\}_{i=1}^r \subset \mathbb{R}^p$ be a set of orthogonal vectors. Then, the set of rank-1 matrices*

$$\mathcal{B} := \left\{ v_i v_i^\top, \, (v_i + v_j)(v_i + v_j)^\top \,\middle|\, 1 \le i < j \le r \right\}$$

*is linearly independent in the space symmetric matrices $\mathsf{Sym}(\mathbb{R}^{p\times p})$.*

Using this construction, the agent can provide feedbacks of the form $(u_i, \sqrt{c_i}y)$ for some $y \in \mathbb{R}^p$ with $\mathbf{\Phi}^*y \neq 0$ and $v_i^\top\mathbf{\Phi}^*v_i = c_i y^\top\mathbf{\Phi}^*y$ to teach the kernel of $\mathbf{\Phi}^*$. For an orthogonal extension $\{u_i\}_{i=r+1}^p$ where $\mathbf{\Phi}^*u_i = 0$ for all $i = r+1, \ldots, p$, feedbacks of the form $(u_i, 0)$ suffice to teach the null space of $\mathbf{\Phi}^*$.

This is the key idea underlying our study on feedback complexity in the general constructive setting that is stated below with the full proof deferred to Appendix F and G.

**Theorem 1** (General Activations)**.** *Let $\mathbf{\Phi}^* \in \mathcal{M}_{\mathsf{F}}$ be a target feature matrix with $\mathrm{rank}(\mathbf{\Phi}^*) = r$. Then, in the setting of constructive feedbacks with general activations, the feedback complexity has a tight bound of $\Theta\left(\frac{r(r+1)}{2} + (p - r) - 1\right)$ for Eq. (1).*

*Proof Outline.* As discussed above we decompose the feature matrix $\mathbf{\Phi}^*$ into its eigenspace and null space, leveraging the linear independence of the constructed feedbacks to ensure that the span covers the necessary orthogonal complements. The upper bound is established with a simple observation: $r(r+1)/2 - 1$ many pairs composed of $\mathcal{B}$ are sufficient to teach $\mathbf{\Phi}^*$ if the null space of $\mathbf{\Phi}^*$ is known, whereas the agent only needs to provide $(p - r)$ many feedbacks corresponding to a basis extension to cover the null space, and hence the stated upper bound is achieved.

The lower bound requires showing that a valid feedback set possesses two spanning properties of $\langle xx^\top - yy^\top \rangle$ for all $(x, y) \in \mathcal{F}$: (1) it must include any $\mathbf{\Phi} \in \mathcal{O}_{\mathbf{\Phi}^*}$ whose column vectors are within the span of eigenvectors of $\mathbf{\Phi}^*$, and (2) it must include any $vv^\top$ for some subset $U$ that spans the null space of $\mathbf{\Phi}^*$ and $v \in U$. $\square$

**Learning with sparse activations** In the discussion above, we demonstrated a strategy for reducing the feedback complexity when general activations are allowed. Now, we aim to understand how this complexity changes when activations are $s$-sparse (see Definition 1) for some $s < p$. Notably, there exists a straightforward construction of rank-1 matrices using a sparse set of activations.

Consider this sparse set of activations $B$ consisting of $\frac{p(p+1)}{2}$ items in $\mathbb{R}^p$ (see (Kumar and Dasgupta, 2024)):

$$B = \{e_i \mid 1 \leq i \leq p\} \cup \{e_i + e_j \mid 1 \leq i < j \leq p\}, \quad (4)$$

where $\{e_i\}$ forms the standard basis. Using a similar argument to Lemma 3, we note that the set of rank-1 matrices

$$\mathcal{B}_{\mathsf{sparse}} := \{uu^\top \mid u \in B\}$$

is linearly independent in the space of symmetric matrices $\mathsf{Sym}(\mathbb{R}^{p \times p})$ and forms a basis. Moreover, every activation in $B_{\mathsf{ext}}$ is at most 2-sparse (see Definition 1). With this, we state the main result on learning with sparse constructive feedback here.

**Theorem 2** (Sparse Activations). *Let $\mathbf{\Phi}^* \in \mathcal{M}_{\mathsf{F}}$ be the target feature matrix. If an agent can construct pairs of activations from a representation space $\mathbb{R}^p$, then the feedback complexity of the feature model $\mathcal{M}_{\mathsf{F}}$ with 2-sparse activations is upper bounded by $\frac{p(p+1)}{2}$.*

*Remark*: While the lower bound from Theorem 1 applies here, sparse settings may require even more feedbacks. Consider a rank-1 matrix $\mathbf{\Phi}^* = vv^\top$ with sparsity$(v) = p$. By the Pigeonhole principle, representing this using $s$-sparse activations requires at least $(p/s)^2$ rank-1 matrices. Thus, for constant sparsity $s = O(1)$, we need $\Omega(p^2)$ feedbacks—implying sparse representation of dense features might not exploit the low-rank structure to minimize feedbacks.

**Algorithm 2** Feature learning with sampled representations

**Given**: Representation space $\mathcal{V} \subset \mathbb{R}^p$, Distribution over representations $\mathcal{D}_\mathcal{V}$, Feature family $\mathcal{M}_{\mathsf{F}}$.

*In batch setting:*

1. Teacher receives sampled representations $\mathcal{V}_n \sim \mathcal{D}_\mathcal{V}$.
2. Teacher picks pairs $\mathcal{F}(\mathcal{V}_n, \mathbf{\Phi}^*) =$
$$\left\{(x, \sqrt{\lambda_x} y) \mid (x, y) \in \mathcal{V}_n^2, \, x^\top \mathbf{\Phi}^* x = \lambda_x \cdot y^\top \mathbf{\Phi}^* y\right\}$$
3. Learner receives $\mathcal{F}$; and obliviously picks a feature matrix $\mathbf{\Phi} \in \mathcal{M}_{\mathsf{F}}$ that satisfy the set of constraints in $\mathcal{F}(\mathcal{V}_n, \mathbf{\Phi}^*)$
4. Learner outputs $\mathbf{\Phi}$.

## 5. Sparse Feature Learning with Sampled Feedback

In general, the assumption of constructive feedback may not hold in practice, as ground truth samples from nature or induced representations of a model are typically independently sampled from the representation space. The literature on Mahalanobis distance learning/dictionary learning has explored distributional assumptions on the sample/activation space (cf (Gribonval et al., 2014)).

In this section, we consider a more realistic scenario where the agent observes a set of representations/activations $\mathcal{V}_n := \{\alpha_1, \alpha_2, \ldots, \alpha_n\} \sim \mathcal{D}_\mathcal{V}$, with $\mathcal{D}_\mathcal{V}$ being an unknown measure over the continuous space $\mathcal{V} \subseteq \mathbb{R}^p$. With these observations, the agent designs pairs of activations to teach a target feature matrix $\mathbf{\Phi}^* \in \mathsf{Sym}_+(\mathbb{R}^{p \times p})$.

As shown in Lemma 2, we can reduce inequality constraints with triplet comparisons to equality constraints with pairs in the constructive setting. However, when the agent is restricted to selecting activations from the sampled set $\mathcal{V}_n$ rather than arbitrarily from $\mathcal{V}$, this reduction no longer holds. Observe that if $\alpha, \beta \sim$ iid $\mathcal{D}_\mathcal{V}$ and $\mathbf{\Phi}^* \neq 0$ a non-degenerate feature matrix, then

$$\alpha^\top \mathbf{\Phi}^* \alpha = \beta^\top \mathbf{\Phi}^* \beta \implies \sum_{i,j} (\alpha_i \alpha_j - \beta_i \beta_j) \mathbf{\Phi}_{ij}^* = 0.$$

This equation represents a non-zero polynomial. According to Sard's Theorem, the zero set of a non-zero polynomial has Lebesgue measure zero. Therefore,

$$\mathcal{P}_{(\alpha, \beta)} \left(\{\alpha^\top \mathbf{\Phi}^* \alpha = \beta^\top \mathbf{\Phi}^* \beta\}\right) = 0.$$

Given this, the agent cannot reliably construct pairs that satisfy the required equality constraints from independently sampled activations. Since a general triplet feedback only provides 3 bits of information, exact recovery up to feature equivalence is impossible. To address these limitations, we consider rescaling the sampled activations to enable the

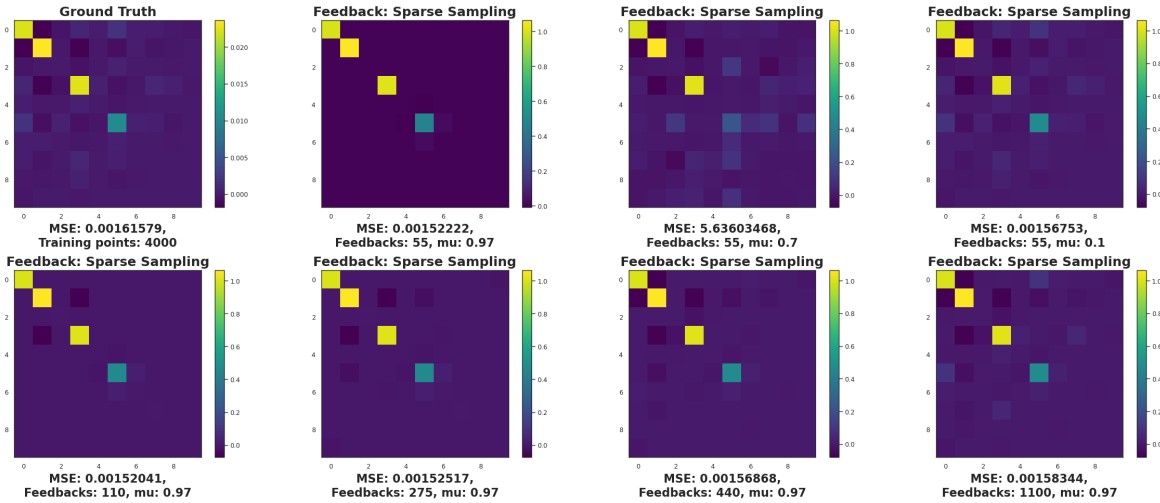

Figure 2: **Sparse sampling.** For the target $f^*(z) = z_0 z_1 z_3 \mathbf{1}(z_5 > 0)$, we apply sparse-sampling feedback with sparsity $mu$ (the probability an entry is zeroed). As $mu$ decreases (denser samples), the canonical complexity $p(p+1)/2 = 55$ suffices to recover $\mathbf{\Phi}^*$. At high sparsity ($mu = 0.97$), more activations—55, 110, ..., 1100—are required to approach ground truth, in agreement with Theorem 4.

agent to design effective pairs for the target feature matrix $\mathbf{\Phi}^* \in \mathcal{M}_F$.

**Rescaled Pairs** For a given matrix $\mathbf{\Phi} \neq 0$, a sampled input $x \sim \mathcal{D}_\mathcal{V}$ is almost never orthogonal, i.e., almost surely $\mathbf{\Phi}x \neq 0$. This property can be utilized to rescale an input and construct pairs that satisfy equality constraints. Specifically, there exist scalars $\gamma, \lambda > 0$ such that (assuming without loss of generality $x^\top \mathbf{\Phi} x > y^\top \mathbf{\Phi} y$),

$$x^\top \mathbf{\Phi} x = \lambda \cdot y^\top \mathbf{\Phi} y + y^\top \mathbf{\Phi} y = (\sqrt{1+\lambda}) y^\top \mathbf{\Phi} (\sqrt{1+\lambda}) y.$$

Thus, the pair $(x, (\sqrt{1+\lambda})y)$ satisfies the equality constraints. With this understanding, we reformulate Algorithm 1 into Algorithm 2. In this section, we analyze the feedback complexity in terms of the minimum number of sampled activations required for the agent to construct an effective feedback set achieving feature equivalence which is illustrated in Fig. 2. Our first result establishes complexity bounds for general activations (without sparsity constraints) sampled from a Lebesgue distribution, with the complete proof provided in Appendix H.

**Theorem 3** (General Sampled Activations). *Consider a representation space $\mathcal{V} \subseteq \mathbb{R}^p$. Assume that the agent receives activations sampled i.i.d from a Lebesgue distribution $\mathcal{D}_\mathcal{V}$. Then, for any target feature matrix $\mathbf{\Phi}^* \in \mathcal{M}_F$, with a tight bound of $n = \Theta\left(\frac{p(p+1)}{2}\right)$ on the feedback complexity, the oblivious learner (almost surely) learns $\mathbf{\Phi}^*$ up to feature equivalence using the feedback set $\mathcal{F}(\mathcal{V}_n, \mathbf{\Phi}^*)$, i.e.,*

$$\mathcal{P}_\mathcal{V} \left( \forall \ \mathbf{\Phi}' \in \mathcal{F}(\mathcal{V}_n, \mathbf{\Phi}^*), \exists \lambda > 0, \mathbf{\Phi}' = \lambda \cdot \mathbf{\Phi}^* \right) = 1.$$

*Proof Outline.* The key observation is that almost surely for any $n \leq p(p+1)/2$ sampled activations on a unit sphere $\mathbb{S}^p$

under Lebesgue measure, the corresponding rank-1 matrices are linearly independent. This is a direct application of Sard's theorem on the zero set of a non-zero polynomial equation, yielding the upper bound. For the lower bound, we use some key necessary properties of a feedback set as elucidated in the proof of Theorem 1. This result essentially fixes activations that need to be spanned by a feedback set, but under a Lebesgue measure on a continuous domain, the probability of sampling a direction is zero. □

We consider a fairly general distribution over sparse activations similar to the signal model in (Gribonval et al., 2015).

**Assumption 1** (Sparse-Distribution). *Each index of a sparse activation vector $\alpha \in \mathbb{R}^p$ is sampled i.i.d from a sparse distribution defined as: for all $i$,*

$$\mathcal{P}(\alpha_i = 0) = p_i, \quad \alpha_i \,|\, \alpha_i \neq 0 \sim Lebesgue((0,1]).$$

With this we state the main theorem of the section with the proof deferred to Appendix I.

**Theorem 4** (Sparse Sampled Activations). *Consider a representation space $\mathcal{V} \subseteq \mathbb{R}^p$. Assume that the agent receives representations sampled i.i.d from a sparse distribution $\mathcal{D}_\mathcal{V}$. Fix a threshold $\delta > 0$, and sparsity parameter $s < p$. Then, for any target feature matrix $\mathbf{\Phi}^* \in \mathcal{M}_F$, with a bound of $n = O\left(p^2 (\frac{2}{p_s^2} \log \frac{2}{\delta})^{1/p^2}\right)$ on the feedback complexity using $s$-sparse feedbacks, the oblivious learner learns $\mathbf{\Phi}^*$ up to feature equivalence with high probability using the feedback set $\mathcal{F}(\mathcal{V}_n, \mathbf{\Phi}^*)$, i.e.,*

$$\mathcal{P}_\mathcal{V} \left( \forall \ \mathbf{\Phi}' \in \mathcal{F}(\mathcal{V}_n, \mathbf{\Phi}^*), \exists \lambda > 0, \mathbf{\Phi}' = \lambda \cdot \mathbf{\Phi}^* \right) \geq (1-\delta),$$

*where $p_s$ depends on $\mathcal{D}_\mathcal{V}$, and sparsity parameter $s$.*

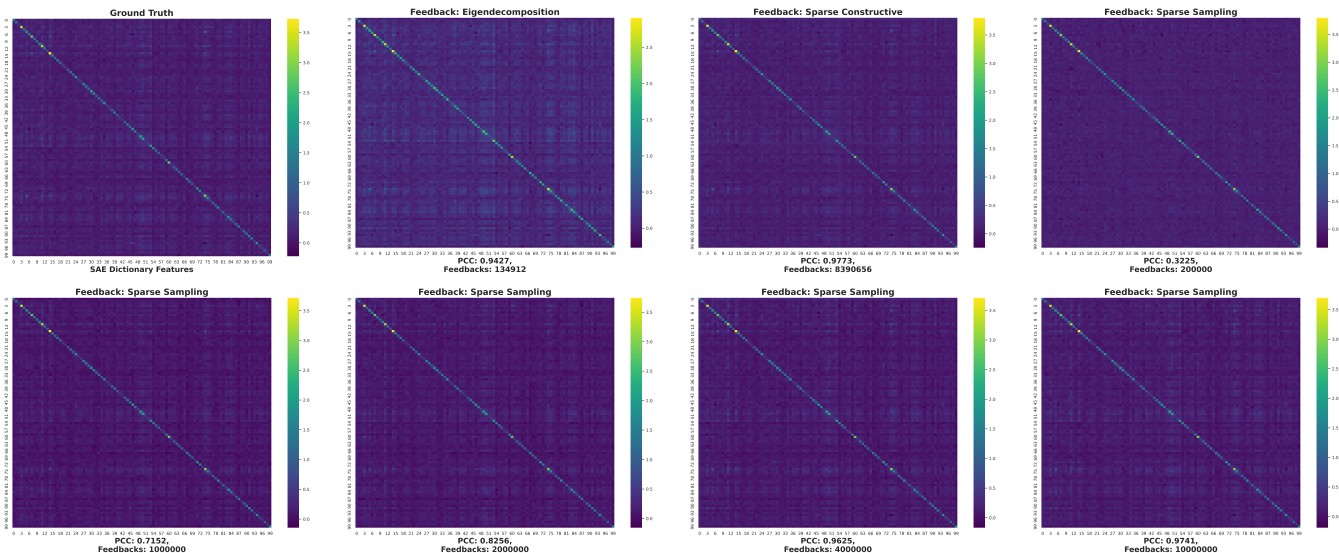

(a) **Visualization of 100 dimensions**: Feature learning on a dictionary retrieved for an MLP layer of ChessGPT of dimension $4096 \times 512$. From left-to-right, top-to-bottom: ground truth SAE, Eigendecomposition (PCC= .9427, 134912 feedbacks), Sparse Constructive (PCC=.9773, 8390656 feedbacks), Sparse Sampling @200000, @1000000, @2000000, @4000000, @10000000 feedbacks.

| Method | Eigendecomp. | Sparse Cons. | Sparse Sampling | | | |
|---|---|---|---|---|---|---|
| | | | 10 M | 4 M | 2 M | 1 M |
| **Feedbacks** | 134912 | 8390656 | 10 M | 4 M | 2 M | 1 M |
| **PCC** | 0.9427 | 0.9773 | 0.9741 | 0.9625 | 0.8256 | 0.7152 |

(b) Pearson correlation coefficient and total feedback count for each method on the same SAE dictionary.

Figure 3: Top: Feature-recovery quality as a function of feedback for a dictionary (of dimension $4096 \times 512$) from an SAE trained for ChessGPT. Bottom: numeric PCC and feedback for each method. Sparse constructive achieves almost perfect correlation (0.9773) in only $\approx 8.4$M queries; sampling with smaller feedback sizes struggle until $\gtrsim 4$M samples.

*Proof Outline.* Using the formulation of Eq. (2), we need to estimate the number of activations the agent needs to receive/sample *before* an induced set of $p(p+1)/2$ many rank-1 linearly independent matrices are found. To estimate this, first we generalize the construction of the set $\mathcal{B}$ from the proof of Theorem 2 to

$$\widehat{U}_g = \left\{ \lambda_i^2 e_i^{\otimes 2} : i \in [p] \right\} \cup \left\{ (\lambda_{iji} e_i + \lambda_{ijj} e_j)^{\otimes 2} : i < j \in [p] \right\}$$

We then analyze a design matrix $\mathbb{M}$ of rank-1 matrices from sampled activations and compute the probability of finding columns with entries semantically similar to those in $\widehat{U}_g$, ensuring a non-trivial determinant. The quantity $p_s$ is the probability that a pattern of these columns is sampled with sparsity at most $s$. The final complexity bound is derived using Hoeffding's inequality and Sterling's approximation. □

## 6. Experimental Setup

We empirically validate our theoretical framework for learning feature matrices. Our sexperiments examine different feedback mechanisms and teaching strategies across both synthetic tasks and large-scale neural networks.

**Feedback Methods:** We evaluate four feedback mechanisms: (1) *Eigendecomposition* uses Lemma 3 to construct feedback based on $\mathbf{\Phi}$'s low rank structure, (2) *Sparse Constructive* builds 2-sparse feedbacks using the basis in Eq. (4), (3) *Random Sampling* generates feedbacks spanning $\mathcal{O}_{\mathbf{\Phi}^*}$ from a Lebesgue distribution, and (4) *Sparse Sampling* creates feedbacks using $s$-sparse samples drawn from a sparse distribution (see Definition 1).

**Teaching Agent:** We implement a teaching agent with access to the target feature matrix to enable numerical analysis. The agent constructs either specific basis vectors or receives activations from distributions (Lebesgue or Sparse) based on the chosen feedback method. For problems with small dimensions, we utilize the cvxpy package to solve constraints of the form $\{\alpha\alpha^\top - yy^\top\}$. When handling larger dimensional features ($5000 \times 5000$), where constraints scale to millions ($p(p+1)/2 \approx 12.5M$), we employ batch-wise gradient descent for matrix regression.

**Features via RFM:** RFM (Radhakrishnan et al., 2024) considers a trainable kernel $K_{\mathbf{\Phi}} : \mathcal{X} \times \mathcal{X} \rightarrow \mathbb{R}$ corresponding to a symmetric, PSD matrix $\mathbf{\Phi}$. At each

---

**Algorithm 3** Optimization via Gradient Descent

---

1. Given a dictionary $U \in \mathbb{R}^{p \times r}$, minimize the loss $\mathcal{L}(U) := \mathcal{L}_{\text{MSE}}(U) + \mathcal{L}_{\text{reg}}(U)$ : where MSE loss and regularization term are:

$$\mathcal{L}_{\text{MSE}}(U) = \frac{1}{|B|} \sum_{i \in B} (\|U \cdot u_i\|^2 - c_i \|U \cdot y\|^2)^2, \ \mathcal{L}_{\text{reg}}(U) = \lambda \|U\|_F^2$$

where $B$ represents the batch of samples, $\lambda = 10^{-4}$ is the regularization coefficient, and $y = e_1$ is the fixed unit vector.

2. For each batch containing indices $i$, values $v$, and targets $c$:

   (a) Construct sparse vectors $u_i$ using $(i, v)$ pairs
   (b) Compute projections: $U^\top u_i$ and $U^\top y$ where $y = e_1$
   (c) Calculate residuals: $r_i = \|U^\top u_i\|^2 - c_i \|U^\top y\|^2$

3. Update $U$ using Adam optimizer with gradient clipping

4. Enforce fixed entries in $U$ after each update ($U[0, 0] = 1$ is enforced to be 1.)

---

iteration, the matrix $\mathbf{\Phi}_t$ is updated for the classifier $f_{\mathbf{\Phi}}(z) = \sum_{y_i \in \mathcal{D}_{\text{train}}} a_i K_{\mathbf{\Phi}}(y_i, z)$ as follows: $\mathbf{\Phi}_{t+1} = \sum_{z \in \mathcal{D}_{\text{train}}} \left(\frac{\partial f_{\mathbf{\Phi}_t}}{\partial z}\right) \left(\frac{\partial f_{\mathbf{\Phi}_t}}{\partial z}\right)^\top$. We train target functions corresponding to monomials over samples in $\mathbb{R}^{10}$ using 4000 training samples. The feature matrix $\mathbf{\Phi}_t$ obtained after $t$ iterations is used as ground truth against learning with feedbacks. Plots are shown in Fig. 1 and Fig. 2.

**SAE features of Large-Scale Models:** We analyze dictionaries from trained sparse autoencoders on Pythia-70M (Biderman et al., 2023) (see Appendix C) and Board Game Models (Karvonen et al., 2024), with dictionary dimensions of $32k \times 512$ and $4096 \times 512$, respectively. We use the dictionaries corresponding to the SAEs trained for various MLP layers of Board Games models: ChessGPT and OthelloGPT considered in (Karvonen et al., 2024), with dimension $4096 \times 512$. Note that $p(p + 1)/2 \approx 8.3M$. For the experiments, we use 3-sparsity on uniform sparse distributions. We present the plots for ChessGPT in Fig. 3 for different feedback methods. Additionally, we provide a table showing the Pearson Correlation Coefficient between the learned feature matrix and the target $\mathbf{\Phi}^*$ in Table 3b.

**Memory-efficient constraint storage** The high dimensionality of model dictionaries makes storing complete activation indices for each feature prohibitively memory-intensive. We address this by enforcing constant sparsity constraints, limiting activations to a maximum sparsity of 3. This constraint enables efficient storage of large-dimensional arrays while preserving the essential characteristics of the features.

**Computational optimization** To efficiently handle constraint satisfaction at scale, we reformulate the problem as a matrix regression task, as detailed in Algorithm 3. The learner maintains a low-rank decomposition of the feature

matrix $\mathbf{\Phi}$, assuming $\mathbf{\Phi} = UU^\top$, where $U$ represents the learned dictionary. This formulation allows for efficient batch-wise optimization over the constraint set while maintaining feasible memory requirements.

Since there could be numerical issues in computation for these large dictionaries, to compare the learnt dictionaries, we compute the Pearson Correlation Coefficient (PCC) of the trained feature matrix $\mathbf{\Phi}'$ with the target matrix $\mathbf{\Phi}^*$ to show their closeness.

$$\rho(\mathbf{\Phi}', \mathbf{\Phi}^*) = \frac{\sum_{i,j}(\mathbf{\Phi}'_{ij} - \bar{\mathbf{\Phi}}')(\mathbf{\Phi}^*_{ij} - \bar{\mathbf{\Phi}}^*)}{\sqrt{\sum_{i,j}(\mathbf{\Phi}'_{ij} - \bar{\mathbf{\Phi}}')^2}\sqrt{\sum_{i,j}(\mathbf{\Phi}^*_{ij} - \bar{\mathbf{\Phi}}^*)^2}}.$$

Note the highest value of $\rho$ is 1.

## 7. Discussion

### 7.1. Limitations and Future Work

The similarity–based feature-learning framework has some major limitations: the learner observes features only up to a normal transformation, so except under strong coherence assumptions (Lemma 1)—full recovery of the underlying dictionary remains open. A natural next step is to relax *exact* feature equivalence and ask instead for an $\varepsilon$-accurate approximation in Frobenius norm. The complexity bounds derived here already translate to the classical statistical-learning setting, but an intriguing open question is whether the gap between these bounds and practical sample requirements can be tightened, perhaps by exploiting the structural insights developed in this work.

### 7.2. Conclusion

Our theoretical bounds reveal that recovering the feature dictionary of a network layer (or a trained SAE) demands at least quadratic sample–complexity in the ambient dimension, which applies across standard settings, e.g., i.i.d. learning, active learning, or machine teaching. This establishes an expressiveness-versus-recoverability trade-off: the more complex or high-dimensional the dictionary, the more feedback/data is required. The quadratic scaling can, however, be reduced under additional structure—e.g., low-rank assumptions—suggesting that leveraging such structure is essential for efficiency. Empirically, we observe that recovery indeed becomes harder in higher dimensions, while incorporating dimensionality-reduction techniques substantially improves performance, motivating future work along these lines. Our results complement the *Neural Feature Ansatz* (Radhakrishnan et al. (2024)) by clarifying when efficient feature recovery is possible: if task-relevant directions lie in a low-dimensional subspace, the required feedback can be sharply reduced. This insight also informs model-distillation, suggesting that smaller students can inherit features efficiently when such a low-rank structure is present.

## Impact Statement

"This paper presents work whose goal is to advance the field of Machine Learning. There are many potential societal consequences of our work, none which we feel must be specifically highlighted here."

## Acknowledgments

Author thanks anonymous reviewers for insightful reviews which helped revise the work in a better context. Author thanks the National Science Foundation for support under grant IIS-2211386 in the duration of this project. Author thanks Sanjoy Dasgupta (UCSD) for helping to develop the preliminary ideas of the work. Author thanks Geelon So (UCSD) for many extended discussions on the project. Author also thanks Mikhail (Misha) Belkin (UCSD) and Enric Boix-Adserà (MIT) for helpful discussion during a visit to the Simons Insitute (UC Berkeley). The general idea of writing was developed while the author was visiting the Simons Institute (UC Berkeley) for a workshop for which the travel was supported by the National Science Foundation (NSF) and the Simons Foundation for the Collaboration on the Theoretical Foundations of Deep Learning through awards DMS-2031883 and #814639.

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

# A. Table of Contents

Here, we provide the table of contents for the appendix of the supplementary.

# B. Notations

Here we provide the glossary of notations followed in the supplementary material.

| Symbol | Description |
|---|---|
| $\alpha, \beta, x, y, z$ | Activations |
| $\mathsf{col}(\boldsymbol{\Phi})$ | Set of columns of matrix $\boldsymbol{\Phi}$ |
| $\mathcal{D}, \mathcal{D}_{\mathsf{sparse}}$ | Distributions over activations |
| $d$ | Dimension of ground-truth sample space |
| $\mathsf{D}$ | Dictionary matrix |
| $\gamma, \lambda, \gamma_i, \lambda_i$ | Eigenvalues of a matrix |
| $\langle \boldsymbol{\Phi}', \boldsymbol{\Phi} \rangle$ | Element-wise product (inner product) of matrices |
| $\mathrm{Ker}(\boldsymbol{\Phi})$ | Kernel of matrix $\boldsymbol{\Phi}$ |
| $\mu_i, u_i, v_i$ | Eigenvectors (orthogonal vectors) |
| $\mathrm{null}(\boldsymbol{\Phi})$ | Null set of matrix $\boldsymbol{\Phi}$ |
| $\mathcal{O}_{\boldsymbol{\Phi}^*}$ | Orthogonal complement of $\boldsymbol{\Phi}^*$ in $\mathsf{Sym}(\mathbb{R}^{p \times p})$ |
| $p$ | Dimension of representation space |
| $\boldsymbol{\Phi}, \Sigma$ | Feature matrix |
| $\boldsymbol{\Phi}_{ij}$ | Entry at $i$th row and $j$th column of $\boldsymbol{\Phi}$ |
| $\boldsymbol{\Phi}^*$ | Target feature matrix |
| $r$ | Rank of a feature matrix |
| $\mathsf{Sym}(\mathbb{R}^{p \times p})$ | Space of symmetric matrices |
| $\mathsf{Sym}_+(\mathbb{R}^{p \times p})$ | Space of PSD, symmetric matrices |
| $\mathsf{VS}(\mathcal{F}, \mathcal{M}_{\mathsf{F}})$ | Version space of $\mathcal{M}_{\mathsf{F}}$ wrt feedback set $\mathcal{F}$ |
| $V_{[r]}$ | The set $\{v_1, v_2, \ldots, v_r\}$ |
| $V_{[p-r]}$ | The set $\{v_{r+1}, \ldots, v_p\}$ |
| $V_{[p]}$ | Complete orthonormal basis $\{v_1, v_2, \ldots, v_p\}$ |
| $\mathcal{V} \subset \mathbb{R}^p$ | Activation/Representation space |
| $\mathcal{X} \subset \mathbb{R}^d$ | Ground truth sample space |

# C. Additional Experiments

In Section 6, we provided details of our experimental setup. In this appendix, we will show the results for some additional experiments: 1) Large-scale SAEs trained on Pythia-70M (Biderman et al., 2023), and 2) extensive experimental results (in Appendix C.1) on a synthetic task as considered in Fig. 1 and Fig. 2.

**Dictionary features of Pythia-70M**  We use the publicly available repository for dictionary learning via sparse autoencoders on neural network activations (Marks et al., 2024a). We consider the dictionaries trained for Pythia-70M (Biderman et al., 2023) (a general-purpose LLM trained on publicly available datasets). We retrieve the corresponding autoencoders for the attention output layers, which have dimensions $32768 \times 512$. Note that $p(p+1)/2 \approx, 512M$. For the experiments, we use 3-sparsity on uniform sparse distributions. We present the plots for ChessGPT in two parts in Fig. 4 and Fig. 5 for different feedback methods.

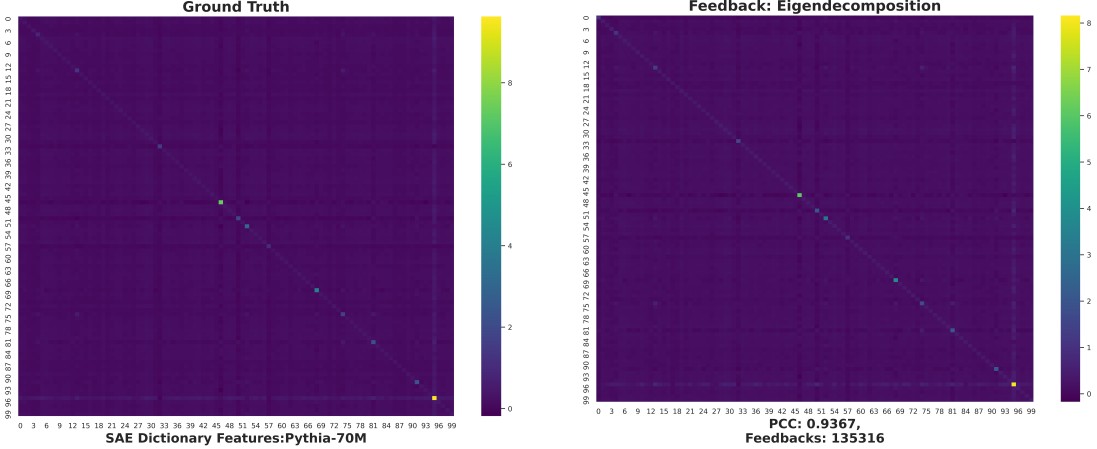

Figure 4: Feature learning on a subsampled dictionary of dimension $4500 \times 512$ of SAE trained for Pythia-70M. Theorem 1 states that Eigendecompostion method requires 135316 constructive feedback. After a few 100 iterations of gradient descent as shown in Algorithm 3, a PCC of 93% is achieved on ground truth. For visualization, only the first 100 dimensions are used.

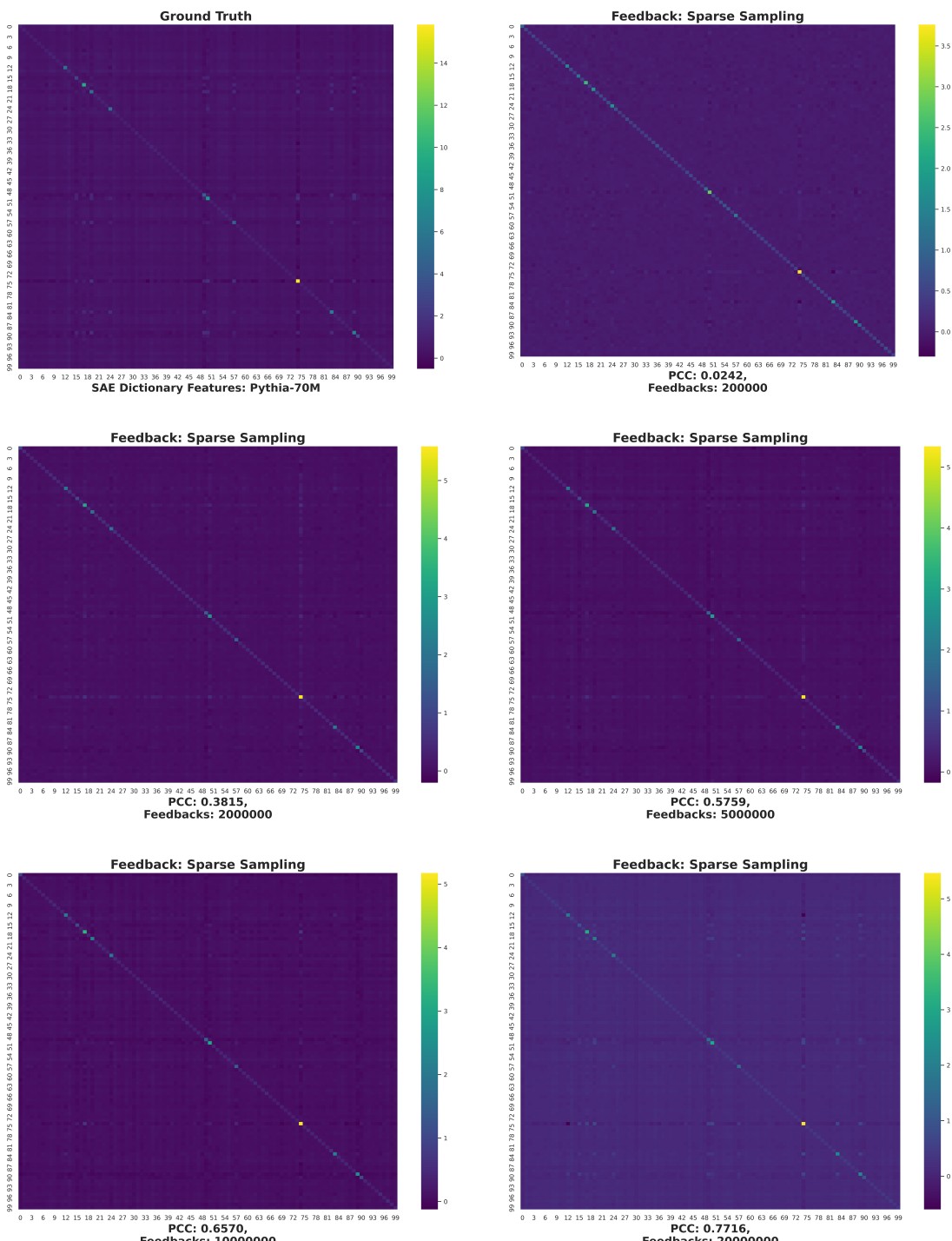

Figure 5: **Sparse sampling for Pythia-70M**: Dimension of feature matrix: $32768 \times 512$ and the rank is 215. Plots for varying feedback complexity sizes. Note that $p(p+1)/2 \approx 512M$. We run experiments with 3-sparse activations for uniform sparse distributions. The Pearson Correlation Coefficient (PCC) to feedback size (PCC, Feedback size) improves as follows: $(200k, .0242), (2M, .38), (5M, .54), (10M, .65),$ and $(20M, .77)$.

## C.1. Verification of theoretical results on a synthetic task

To validate our theoretical results, we compare the upper bounds derived in Theorem 1–4 against empirical performance on a controlled synthetic task. This experiment aims to assess how tightly the theoretical feedback complexity aligns with the actual number of feedback queries required to achieve feature recovery up to linear scaling equivalence (Definition 2).

We consider a monomial regression task defined by

$$y = f^*(\boldsymbol{x}) = \boldsymbol{x}_1 \boldsymbol{x}_2 \boldsymbol{x}_3 \boldsymbol{x}_4 \cdot \mathbf{1}(\boldsymbol{x}_5 > 0),$$

which induces a target feature matrix $\boldsymbol{\Phi}^*$ (as constructed by the Recursive Feature Machine (Radhakrishnan et al., 2024), see Section 6).

**Setup.** Inputs $\boldsymbol{x} \in \mathbb{R}^{10}$ are sampled from a Gaussian distribution $\mathcal{N}(0, 0.5\mathbb{I}_{10})$. We train an RFM classifier on 5000 training samples to obtain $\boldsymbol{\Phi}^*$, and the teaching agent has access to this feature matrix for generating feedback.

We evaluated the following four feedback mechanisms: Eigendecomposition, Sparse Constructive, Random Sampling, and Sparse Sampling (Section 4 and Section 5).

For each method, we report:

1. The number of feedbacks provided.

2. The empirical mean squared error (MSE) compared to the target MSE achieved using $\boldsymbol{\Phi}^*$.

3. The theoretical upper bound on the number of feedbacks.

**Theoretical vs Empirical Observations.** The target feature matrix $\boldsymbol{\Phi}^*$ has rank $r = 8$, and the ambient input dimension is $p = 10$, giving $p(p+1)/2 = 55$ as the total number of degrees of freedom used in the stated bounds.

- **Eigendecomposition**: Theoretical bound (Theorem 1) is $\frac{r(r+1)}{2} + p - r = 38$. As shown in Figure 6, this exact number of feedbacks is sufficient to match the target MSE (mean squared error) empirically.

- **Sparse Constructive**: Using 2-sparse feedbacks (Theorem 2), the theoretical bound remains 55. As illustrated in Figure 6, the empirical performance saturates at the target MSE within this bound.

- **Random Sampling**: Feedback is sampled uniformly at random. We evaluate empirical performance at 20%, 30%, 50%, 70%, and 100% of the theoretical bound: 55 (as computed using Theorem 3), as shown in Figure 6. The gradual reduction in MSE confirms that the learning curve aligns well with the theoretical complexity.

  *Remark*: Given that these are sampled runs (not averaged), in some cases, the MSE might be higher even if the feedback set is increased (implying that an increase in feedback didn't lead to relevant independent directions). But, averaging over runs, we note that the MSE gradually reduces in MSE with the stated theoretical bound.

- **Sparse Sampling**: Our first experiment is for 4-sparse activations in Figure 7, where each coordinate is nonzero with probability $1 - \mu = 0.2$, and the nonzero values are drawn from $\mathcal{U}(0, 1)$. Using a success threshold of $\delta = 0.05$, Theorem 4 yields a bound of 117 feedbacks. Figure 7 shows MSE values at multiples (30% to 2000%) of the total number of degrees of freedom (55). As expected, MSE converges to the target MSE once the feedback size reaches the theoretical threshold.

  We perform several experiments with different values of $\delta, \mu$, and sparsity level as shown in Figure 8-11.

  *Remark*: Since the bounds are independent of the distribution of a coordinate being non-zero, the bounds don't change even if we use a distribution other than the uniform distribution.

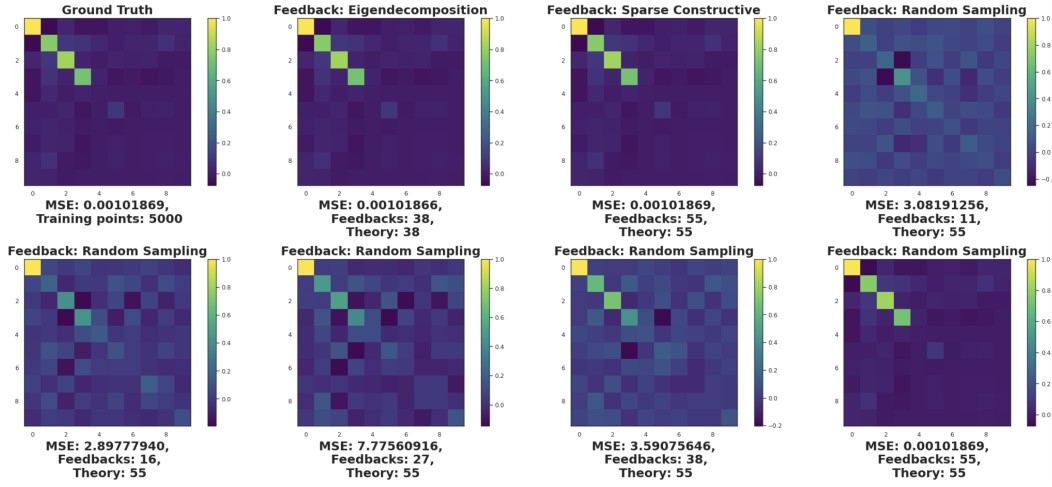

Figure 6: Empirical performance for Eigendecomposition, Sparse Constructive, and Random Sampling.

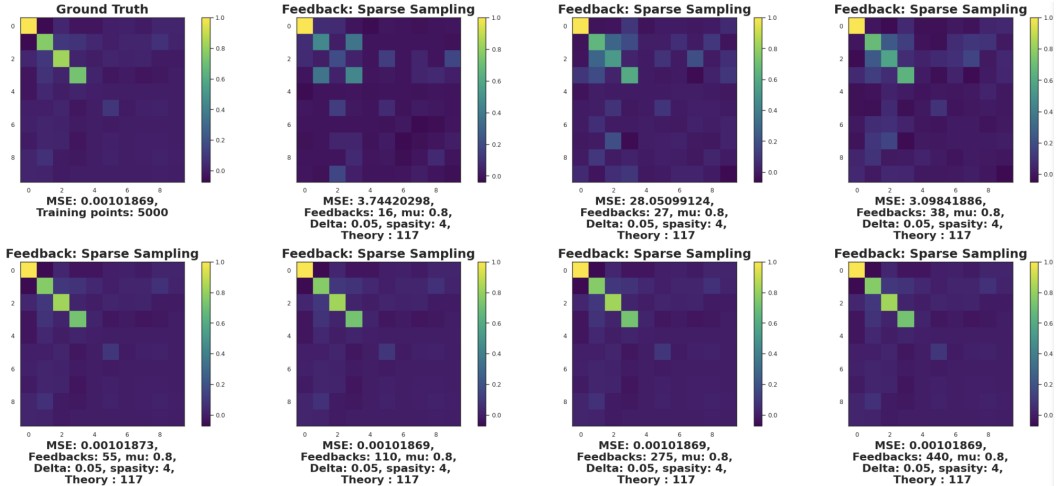

Figure 7: Empirical performance for the Sparse Sampling feedback mechanism.

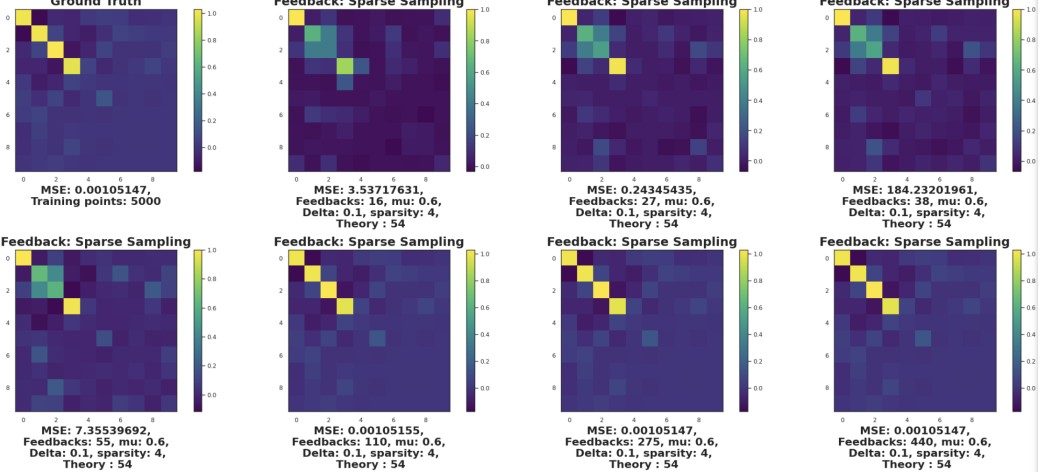

Figure 8: Empirical performance for the Sparse Sampling feedback mechanism.

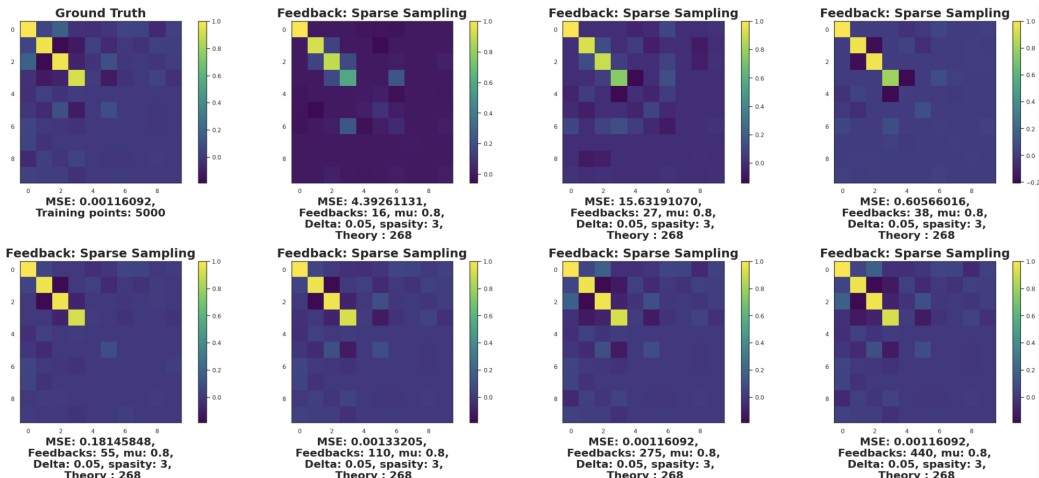

Figure 9: Empirical performance for the Sparse Sampling feedback mechanism.

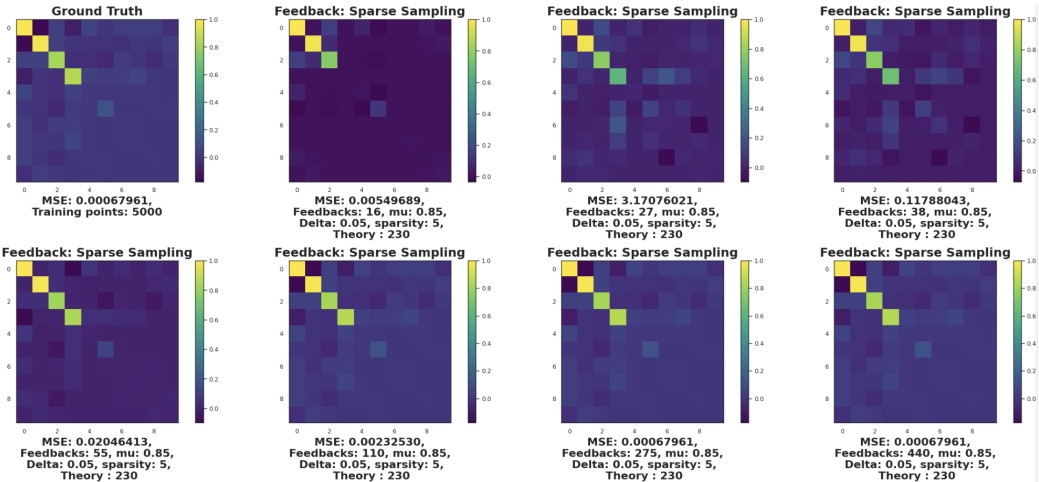

Figure 10: Empirical performance for the Sparse Sampling feedback mechanism.

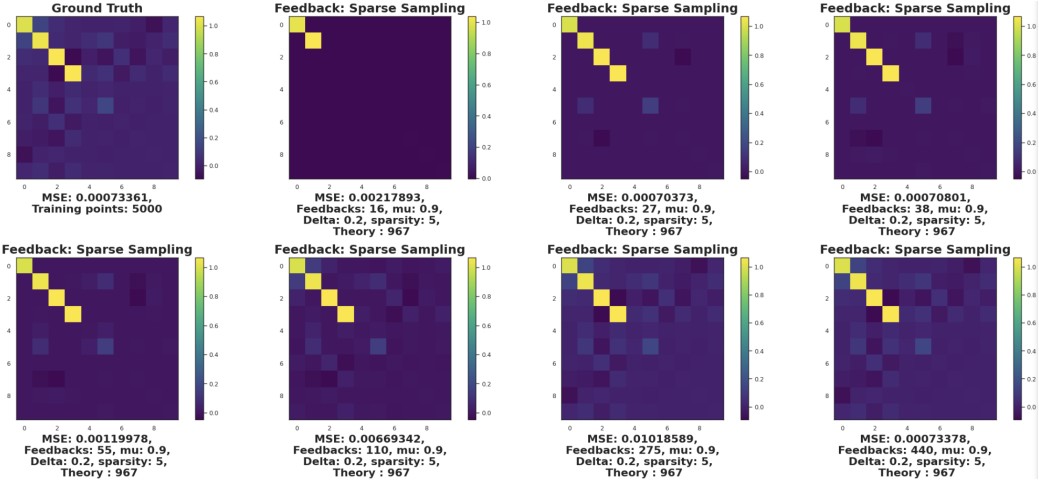

Figure 11: Empirical performance for the Sparse Sampling feedback mechanism.

# D. Proof of Lemma 1

In this appendix we restate and provide the proof of Lemma 1.

**Lemma 1** (Recovering orthogonal atoms). *Let $\boldsymbol{\Phi} \in \mathbb{R}^{p \times p}$ be a symmetric positive semi-definite matrix. Define the set of orthogonal Cholesky decompositions of $\boldsymbol{\Phi}$ as*

$$\mathcal{W}_{CD} = \left\{ \boldsymbol{U} \in \mathbb{R}^{p \times r} \, \middle| \, \boldsymbol{\Phi} = \boldsymbol{U}\boldsymbol{U}^\top \text{ and } \boldsymbol{U}^\top \boldsymbol{U} = diag(\lambda_1, \ldots, \lambda_r) \right\},$$

*where $r = rank(\boldsymbol{\Phi})$ and $\lambda_1, \lambda_2, \ldots, \lambda_r$ are the eigenvalues of $\boldsymbol{\Phi}$ in descending order. Then, for any two matrices $\boldsymbol{U}, \boldsymbol{U}' \in \mathcal{W}_{CD}$, there exists an orthogonal matrix $R \in \mathbb{R}^{r \times r}$ such that*

$$\boldsymbol{U}' = \boldsymbol{U}\boldsymbol{R},$$

*where $\boldsymbol{R}$ is block diagonal with orthogonal blocks corresponding to any repeated diagonal entries $d_i$ in $\boldsymbol{U}^\top \boldsymbol{U}$. Additionally, each column of $\boldsymbol{U}'$ can differ from the corresponding column of $\boldsymbol{U}$ by a sign change.*

*Proof.* Let $\mathbf{U}, \mathbf{U}' \in \mathcal{W}_{CD}$ be two orthogonal Cholesky decompositions of $\boldsymbol{\Phi}$. Define $\mathbf{R} = \mathbf{U}^\top \mathrm{diag}(1/\lambda_1, \ldots, 1/\lambda_r)\mathbf{U}'$. We will show that this matrix satisfies our requirements through the following steps:

First, we show that $\mathbf{R}$ is orthogonal. Note,

$$
\begin{aligned}
\mathbf{R}^\top \mathbf{R} &= (\mathbf{U}^\top \mathrm{diag}(1/\lambda_1, \ldots, 1/\lambda_r)\mathbf{U}')^\top (\mathbf{U}^\top \mathrm{diag}(1/\lambda_1, \ldots, 1/\lambda_r)\mathbf{U}') \\
&= \mathbf{U}'^\top \mathrm{diag}(1/\lambda_1, \ldots, 1/\lambda_r)\mathbf{U}\mathbf{U}^\top \mathrm{diag}(1/\lambda_1, \ldots, 1/\lambda_r)\mathbf{U}' \\
&= \mathbf{U}'^\top \mathrm{diag}(1/\lambda_1, \ldots, 1/\lambda_r)\boldsymbol{\Phi}\mathrm{diag}(1/\lambda_1, \ldots, 1/\lambda_r)\mathbf{U}' \\
&= \mathbf{U}'^\top \mathrm{diag}(1/\lambda_1, \ldots, 1/\lambda_r)\mathbf{U}'\mathbf{U}'^\top \mathrm{diag}(1/\lambda_1, \ldots, 1/\lambda_r)\mathbf{U}' \\
&= \mathbf{U}'^\top \mathrm{diag}(1/\lambda_1, \ldots, 1/\lambda_r)\mathbf{U}' \\
&= \mathbf{I}_r
\end{aligned}
$$

Similarly,

$$
\begin{aligned}
\mathbf{R}\mathbf{R}^\top &= \mathbf{U}^\top \mathrm{diag}(1/\lambda_1, \ldots, 1/\lambda_r)\mathbf{U}'(\mathbf{U}')^\top \mathrm{diag}(1/\lambda_1, \ldots, 1/\lambda_r)\mathbf{U} \\
&= \mathbf{U}^\top \mathrm{diag}(1/\lambda_1, \ldots, 1/\lambda_r)\boldsymbol{\Phi}\mathrm{diag}(1/\lambda_1, \ldots, 1/\lambda_r)\mathbf{U} \\
&= \mathbf{U}^\top \mathrm{diag}(1/\lambda_1, \ldots, 1/\lambda_r)\mathbf{U}\mathbf{U}^\top \mathbf{U} \\
&= \mathbf{U}^\top \mathrm{diag}(1/\lambda_1, \ldots, 1/\lambda_r)\mathbf{U}\mathrm{diag}(\lambda_1, \ldots, \lambda_r) \\
&= \mathbf{I}_r
\end{aligned}
$$

Now we show that $\mathbf{U}' = \mathbf{U}\mathbf{R}$.

$$
\begin{aligned}
\mathbf{U}\mathbf{R} &= \mathbf{U}\mathbf{U}^\top \mathrm{diag}(1/\lambda_1, \ldots, 1/\lambda_r)\mathbf{U}' \\
&= \boldsymbol{\Phi}\mathrm{diag}(1/\lambda_1, \ldots, 1/\lambda_r)\mathbf{U}' \\
&= \mathbf{U}'\mathbf{U}'^\top \mathbf{U}'\mathrm{diag}(1/\lambda_1, \ldots, 1/\lambda_r) \\
&= \mathbf{U}'\mathrm{diag}(\lambda_1, \ldots, \lambda_r)\mathrm{diag}(1/\lambda_1, \ldots, 1/\lambda_r) \\
&= \mathbf{U}'
\end{aligned}
$$

To show that $\mathbf{R}$ is block diagonal with orthogonal blocks corresponding to repeated eigenvalues, consider the partitioning based on distinct eigenvalues. Let $\mathcal{I}_k = \{i \mid \lambda_i = \gamma_k\}$ be the set of indices corresponding to the $k$-th distinct eigenvalue $\gamma_k$ of $\boldsymbol{\Phi}$, for $k = 1, \ldots, K$, where $K$ is the number of distinct eigenvalues. Let $m_k = |\mathcal{I}_k|$ denote the multiplicity of $\gamma_k$.

Define $\mathbf{U}_k$ and $\mathbf{U}'_k$ as the submatrices of $\mathbf{U}$ and $\mathbf{U}'$ consisting of columns indexed by $\mathcal{I}_k$, respectively.

Now, consider the block $\mathbf{R}_{k\ell}$ of $\mathbf{R}$ corresponding to eigenvalues $\gamma_k$ and $\gamma_\ell$. For $k \neq \ell$, $\mathbf{U}_k$ and $\mathbf{U}'_\ell$ correspond to different eigenspaces (as $\gamma_k \neq \gamma_\ell$), and thus their inner product is zero. Hence,

$$\mathbf{U}_k^\top \operatorname{diag}\left(\frac{1}{\lambda_1}, \ldots, \frac{1}{\lambda_r}\right)\mathbf{U}'_\ell = \mathbf{0}_{m_k \times m_\ell}.$$

This implies $\mathbf{R}_{k\ell} = \mathbf{0}_{m_k \times m_\ell}$ for $k \neq \ell$.

But then $\mathbf{R}$ must be block diagonal:

$$\mathbf{R} = \begin{bmatrix} \mathbf{R}_1 & \mathbf{0} & \cdots & \mathbf{0} \\ \mathbf{0} & \mathbf{R}_2 & \cdots & \mathbf{0} \\ \vdots & \vdots & \ddots & \vdots \\ \mathbf{0} & \mathbf{0} & \cdots & \mathbf{R}_K \end{bmatrix},$$

where each $\mathbf{R}_k \in \mathbb{R}^{m_k \times m_k}$ is an orthogonal matrix. For eigenvalues with multiplicity one ($m_k = 1$), the corresponding block $\mathbf{R}_k$ is a $1 \times 1$ orthogonal matrix. The only possibilities are:

$$\mathbf{R}_k = [1] \quad \text{or} \quad \mathbf{R}_k = [-1],$$

representing a sign change in the corresponding column of $\mathbf{U}$. For eigenvalues with multiplicity greater than one ($m_k > 1$), each block $\mathbf{R}_k$ can be any $m_k \times m_k$ orthogonal matrix. This allows for rotations within the eigenspace corresponding to the repeated eigenvalue $\gamma_k$.

Combining all steps, we have shown that:
$$\mathbf{U}' = \mathbf{U}\mathbf{R},$$

where $\mathbf{R}$ is an orthogonal, block-diagonal matrix. Each block $\mathbf{R}_k$ corresponds to a distinct eigenvalue $\gamma_k$ of $\mathbf{\Phi}$ and is either a $1 \times 1$ matrix with entry $\pm 1$ (for unique eigenvalues) or an arbitrary orthogonal matrix of size equal to the multiplicity of $\gamma_k$ (for repeated eigenvalues). This completes the proof of the lemma.

$\square$

## E. Worst-case bounds: Constructive case

In this Appendix, we provide the proof of the lower bound as stated in Proposition 1. Before we prove this lower bound, we state a useful property of the sum of a symmetric, PSD matrix and a general symmetric matrix in $\operatorname{Sym}(\mathbb{R}^{p \times p})$.

**Lemma 5.** *Let $\mathbf{\Phi} \in \operatorname{Sym}_+(\mathbb{R}^{p \times p})$ be a symmetric matrix with full rank, i.e., $\operatorname{rank}(\mathbf{\Phi}) = p$. For any arbitrary symmetric matrix $\mathbf{\Phi}' \in \operatorname{Sym}(\mathbb{R}^{p \times p})$, there exists a positive scalar $\lambda > 0$ such that the matrix $(\mathbf{\Phi} + \lambda\mathbf{\Phi}')$ is positive semidefinite.*

*Proof.* Since $\mathbf{\Phi}$ is symmetric and has full rank, it admits an eigendecomposition:

$$\mathbf{\Phi} = \sum_{i=1}^{p} \lambda_i u_i u_i^\top,$$

where $\{\lambda_i\}_{i=1}^{p}$ are the positive eigenvalues and $\{u_i\}_{i=1}^{p}$ are the corresponding orthonormal eigenvectors of $\mathbf{\Phi}$.

Define the constant $\gamma$ as the maximum absolute value of the quadratic forms of $\mathbf{\Phi}'$ with respect to the eigenvectors of $\mathbf{\Phi}$:

$$\gamma := \max_{1 \leq i \leq p} \left| u_i^\top \mathbf{\Phi}' u_i \right|.$$

Let $\lambda$ be chosen as:

$$\lambda := \frac{\min_{1 \leq i \leq p} \lambda_i}{\gamma}.$$

For each eigenvector $u_i$, consider the quadratic form of $(\mathbf{\Phi} + \lambda\mathbf{\Phi}')$:

$$u_i^\top(\mathbf{\Phi} + \lambda\mathbf{\Phi}')u_i = \lambda_i + \lambda u_i^\top \mathbf{\Phi}' u_i \geq \lambda_i - \lambda\gamma = \lambda_i - \frac{\min \lambda_i}{\gamma}\gamma = \lambda_i - \min \lambda_i \geq 0.$$

This shows that each eigenvector $u_i$ satisfies:

$$u_i^\top(\mathbf{\Phi} + \lambda\mathbf{\Phi}')u_i \geq 0.$$

Since $\{u_i\}_{i=1}^p$ forms an orthonormal basis for $\mathbb{R}^p$, for any vector $x \in \mathbb{R}^p$, we can express $x$ as $x = \sum_{i=1}^p a_i u_i$. Then:

$$x^\top(\mathbf{\Phi} + \lambda\mathbf{\Phi}')x = \sum_{i=1}^p a_i^2 u_i^\top(\mathbf{\Phi} + \lambda\mathbf{\Phi}')u_i \geq 0,$$

since each term in the sum is non-negative.

Therefore, $(\mathbf{\Phi} + \lambda\mathbf{\Phi}')$ is positive semidefinite. $\qquad\square$

Now, we provide the proof of Proposition 1 in the following:

*Proof of Proposition 1.* Assume, for contradiction, that there exists a feedback set $\mathcal{F}(\mathcal{V}, \mathcal{M}_\mathsf{F}, \mathbf{\Phi}^*)$ for Eq. (1) with size $|\mathcal{F}| < \left(\frac{p(p+1)}{2} - 1\right)$.

For each pair $(y, z) \in \mathcal{F}$, $\mathbf{\Phi}^*$ is orthogonal to $(yy^\top - zz^\top)$, implying that $(yy^\top - zz^\top) \in \mathcal{O}_{\mathbf{\Phi}^*}$, the orthogonal complement of $\mathbf{\Phi}^*$. Therefore,

$$\text{span}\left\langle\{yy^\top - zz^\top\}_{(y,z)\in\mathcal{F}}\right\rangle \subset \mathcal{O}_{\mathbf{\Phi}^*}.$$

This leads to

$$\mathbf{\Phi}^* \perp \text{span}\left\langle\{yy^\top - zz^\top\}_{(y,z)\in\mathcal{F}}\right\rangle.$$

Since $|\mathcal{F}| < \frac{p(p+1)}{2} - 1$, we have

$$\dim\left(\text{span}\left\langle\{yy^\top - zz^\top\}\right\rangle\right) < \frac{p(p+1)}{2} - 1.$$

Adding $\mathbf{\Phi}^*$ to this span increases the dimension by at most one:

$$\dim\left(\text{span}\left\langle\mathbf{\Phi}^* \cup \{yy^\top - zz^\top\}_{(y,z)\in\mathcal{F}}\right\rangle\right) \leq \frac{p(p+1)}{2} - 1.$$

Since $\mathsf{Sym}(\mathbb{R}^{p\times p})$ is a vector space with $\dim(\mathsf{Sym}(\mathbb{R}^{p\times p})) = \frac{p(p+1)}{2}$, there exists a symmetric matrix $\mathbf{\Phi}' \in \mathcal{O}_{\mathbf{\Phi}^*}$ such that

$$\mathbf{\Phi}' \perp (yy^\top - zz^\top) \quad \forall (y, z) \in \mathcal{F}.$$

By Lemma 5, there exists $\lambda > 0$ such that $\mathbf{\Phi}^* + \lambda\mathbf{\Phi}'$ is PSD and symmetric. Since $\mathbf{\Phi}' \in \mathcal{O}_{\mathbf{\Phi}^*}$ and $\mathbf{\Phi}'$ is not a scalar multiple of $\mathbf{\Phi}^*$, the matrix $\mathbf{\Phi}^* + \lambda\mathbf{\Phi}'$ is not related to $\mathbf{\Phi}^*$ via linear scaling. However, it still satisfies Eq. (1), contradicting the minimality of $\mathcal{F}$.

Thus, any feedback set must satisfy

$$|\mathcal{F}| \geq \frac{p(p+1)}{2} - 1.$$

This establishes the stated lower bound on the feedback complexity of the feedback set. $\qquad\square$

# F. Proof of Theorem 1: Upper bound

Below we provide proof of the upper bound stated in Theorem 1.

Consider the eigendecomposition of the matrix $\mathbf{\Phi}^*$. There exists a set of orthonormal vectors $\{v_1, v_2, \ldots, v_r\}$ with corresponding eigenvalues $\{\gamma_1, \gamma_2, \ldots, \gamma_r\}$ such that

$$\mathbf{\Phi}^* = \sum_{i=1}^{r} \gamma_i v_i v_i^\top \tag{5}$$

Denote the set of orthogonal vectors $\{v_1, v_2, \ldots, v_r\}$ as $V_{[r]}$.

Let $\{v_{r+1}, \ldots, v_p\}$, denoted as $V_{[p-r]}$, be an orthogonal extension to the vectors in $V_{[r]}$ such that

$$V_{[r]} \cup V_{[p-r]} = \{v_1, v_2, \ldots, v_p\}$$

forms an orthonormal basis for $\mathbb{R}^p$. Denote the complete basis $\{v_1, v_2, \ldots, v_p\}$ as $V_{[p]}$.

Note that $\{v_{r+1}, \ldots, v_p\}$ precisely defines the null space of $\mathbf{\Phi}^*$, i.e.,

$$\text{null}(\mathbf{\Phi}^*) = \text{span} \langle \{v_{r+1}, \ldots, v_p\} \rangle.$$

The key idea of the proof is to manipulate this null space to satisfy the feedback set condition in Eq. (2) for the target matrix $\mathbf{\Phi}^*$. Since $\mathbf{\Phi}^*$ has rank $r \leq p$, the number of degrees of freedom is exactly $\frac{r(r+1)}{2}$. Alternatively, the span of the null space of $\mathbf{\Phi}^*$, which has dimension exactly $p - r$, fixes the remaining entries in $\mathbf{\Phi}^*$.

Using this intuition, the teacher can provide pairs $(y, z) \in \mathcal{V}^2$ to teach the null space and the eigenvectors $\{v_1, v_2, \ldots, v_r\}$ separately. However, it is necessary to ensure that this strategy is optimal in terms of sample efficiency. We confirm the optimality of this strategy in the next two lemmas.

## F.1. Feedback set for the null space of $\mathbf{\Phi}^*$

Our first result is on nullifying the null set of $\mathbf{\Phi}^*$ in the Eq. (2). Consider a partial feedback set

$$\mathcal{F}_{\text{null}} = \{(0, v_i)\}_{i=r+1}^{p}$$

**Lemma 6.** *If the teacher provides the set $\mathcal{F}_{\text{null}}$, then the null space of any PSD symmetric matrix $\mathbf{\Phi}'$ that satisfies Eq. (2) contains the span of $\{v_{r+1}, \ldots, v_p\}$, i.e.,*

$$\{v_{r+1}, \ldots, v_p\} \subseteq \text{null}(\mathbf{\Phi}').$$

*Proof.* Let $\mathbf{\Phi}' \in \text{Sym}_+(\mathbb{R}^{p \times p})$ be a matrix that satisfies Eq. (2) (note that $\mathbf{\Phi}^*$ satisfies Eq. (2)). Thus, we have the following equality constraints:

$$\forall (0, v) \in \mathcal{F}_{\text{null}}, \quad v^\top \mathbf{\Phi}' v = 0.$$

Since $\{v_{r+1}, \ldots, v_p\}$ is a set of linearly independent vectors, it suffices to show that

$$\forall v \in V_{[d-r]}, \quad v^\top \mathbf{\Phi}' v = 0 \implies \mathbf{\Phi}' v = 0. \tag{6}$$

To prove Eq. (6), we utilize general properties of the eigendecomposition of a symmetric, positive semi-definite matrix. We express $\mathbf{\Phi}'$ in its eigendecomposition as

$$\mathbf{\Phi}' = \sum_{i=1}^{s} \gamma_i' u_i u_i^\top,$$

where $\{u_i\}_{i=1}^{s}$ are the eigenvectors and $\{\gamma_i'\}_{i=1}^{s}$ are the corresponding eigenvalues of $\mathbf{\Phi}'$. Assume that $x \neq 0 \in \mathbb{R}^p$ satisfies

$$x^\top \mathbf{\Phi}' x = 0.$$

Consider the decomposition $x = \sum_{i=1}^{s} a_i u_i + v'$ for scalars $a_i$ and $v' \perp \{u_i\}_{i=1}^{s}$. Now, expanding the equation above, we get

$$
\begin{aligned}
x^\top \mathbf{\Phi}' x &= \left( \sum_{i=1}^{s} a_i u_i + v' \right)^\top \mathbf{\Phi}' \left( \sum_{i=1}^{s} a_i u_i + v' \right) \\
&= \left( \sum_{i=1}^{s} a_i u_i \right)^\top \mathbf{\Phi}' \left( \sum_{i=1}^{s} a_i u_i \right) + v'^\top \mathbf{\Phi}' \left( \sum_{i=1}^{s} a_i u_i \right) + \left( \sum_{i=1}^{s} a_i u_i \right) \mathbf{\Phi}' v' + v'^\top \mathbf{\Phi}' v' \\
&= \left( \sum_{i=1}^{s} a_i u_i \right)^\top \left( \sum_{i=1}^{s} \gamma_i' u_i u_i^\top \right) \left( \sum_{i=1}^{s} a_i u_i \right) + \underbrace{2 v'^\top \left( \sum_{i=1}^{s} \gamma_i' u_i u_i^\top \right) \left( \sum_{i=1}^{s} a_i u_i \right) + v'^\top \left( \sum_{i=1}^{s} \gamma_i' u_i u_i^\top \right) v'}_{= 0 \text{ as } v' \perp \{u_i\}} \\
&= \sum_{i,j,k} a_i u_i^\top (\gamma_j' u_j u_j^\top) a_k u_k \\
&= \sum_{i=1}^{s} a_i^2 \gamma_i' = 0
\end{aligned}
$$

Since $\gamma_i' > 0$ for all $i = 1, \ldots, s$ (because $\mathbf{\Phi}'$ is PSD), it follows that each $a_i = 0$. Therefore,

$$
\mathbf{\Phi}' x = \mathbf{\Phi}' v' = 0.
$$

This implies that $x \in \text{null}(\mathbf{\Phi}')$, thereby proving Eq. (6).

Hence, if the teacher provides $\mathcal{F}_{\text{null}}$, any solution $\mathbf{\Phi}'$ to Eq. (2) must satisfy

$$
\{v_{r+1}, \ldots, v_p\} \subseteq \text{null}(\mathbf{\Phi}').
$$

$\square$

With this we will argue that the feedback setup in Eq. (2) can be decomposed in two parts: first is teaching the null set $\text{null}(\mathbf{\Phi}^*) := \text{span} \left\langle \{v_i\}_{i=r+1}^{n} \right\rangle$, and second is teaching $\mathcal{S}_{\mathbf{\Phi}^*} = \text{span} \left\langle \{v_i\}_{i=1}^{r} \right\rangle$ in the form of $\mathbf{\Phi}^* = \sum_{i=1}^{r} \gamma_i v_i v_i^\top$.

Lemma 6 implies that using a feedback set of the form $\mathcal{F}_{\text{null}}$ any solution $\mathbf{\Phi}' \in \text{Sym}_+(\mathbb{R}^{p \times p})$ to Eq. (2) satisfies the property $V_{[d-r]} \subset \text{null}(\mathbf{\Phi}')$. Furthermore, $|\mathcal{F}_{\text{null}}| = p - r$.

### F.2. Feedback set for the kernel of $\mathbf{\Phi}^*$

Next, we discuss how to teach $V_{[r]}$, i.e. $V_{[r]}$ span the rows of any solution $\mathbf{\Phi}' \in \text{Sym}_+(\mathbb{R}^{p \times p})$ to Eq. (2) with the corresponding eigenvalues $\{\gamma_i\}_{i=1}^{r}$. We show that if the search space of metrics in Eq. (2) is the version space $\text{VS}(\mathcal{M}_F, \mathcal{F}_{\text{null}})$ which is a restriction of the space $\mathcal{M}_F$ to feedback set $\mathcal{F}_{\text{null}}$, then a feedback set of size at most $\frac{r(r+1)}{2} - 1$ is sufficient to teach $\mathbf{\Phi}^*$ up to feature equivalence. Thus, we consider the reformation of the problem in Eq. (2) as

$$
\forall (y, z) \in \mathcal{F}(\mathcal{X}, \text{VS}(\mathcal{M}_F, \mathcal{F}_{\text{null}}), \mathbf{\Phi}^*), \quad \mathbf{\Phi} \cdot (yy^\top - zz^\top) = 0 \tag{7}
$$

where the feedback set $\mathcal{F}(\mathcal{X}, \text{VS}(\mathcal{M}_F, \mathcal{F}_{\text{null}}), \mathbf{\Phi}^*)$ is devised to solve a smaller space $\text{VS}(\mathcal{M}_F, \mathcal{F}_{\text{null}}) := \{\mathbf{\Phi} \in \mathcal{M}_F \mid \mathbf{\Phi} v = 0, \forall (0, v) \in \mathcal{F}_{\text{null}}\}$. With this state the following useful lemma on the size of the restricted feedback set $\mathcal{F}(\mathcal{X}, \text{VS}(\mathcal{M}_F, \mathcal{F}_{\text{null}}), \mathbf{\Phi}^*)$.

**Lemma 7.** *Consider the problem as formulated in Eq. (7) in which the null set $\text{null}(\mathbf{\Phi}^*)$ of the target matrix $\mathbf{\Phi}^*$ is known. Then, the teacher sufficiently and necessarily finds a set $\mathcal{F}(\mathcal{X}, \text{VS}(\mathcal{F}_{\text{null}}), \mathbf{\Phi}^*)$ of size $\frac{r(r+1)}{2} - 1$ for oblivious learning up to feature equivalence.*

*Proof.* Note that any solution $\mathbf{\Phi}'$ of Eq. (7) has its columns spanned exactly by $V_{[r]}$. Alternatively, if we consider the eigendecompostion of $\mathbf{\Phi}'$ then the corresponding eigenvectors exists in $span \left\langle V_{[r]} \right\rangle$. Furthermore, note that $\mathbf{\Phi}^*$ is of rank $r$ which implies there are only $\frac{r(r+1)}{2}$ degrees of freedom, i.e. entries in the matrix $\mathbf{\Phi}^*$, that need to be fixed.

Thus, there are exactly $r$ linearly independent columns of $\boldsymbol{\Phi}^*$, indexed as $\{j_1, j_2, \ldots, j_r\}$. Now, consider the set of matrices

$$\left\{ \boldsymbol{\Phi}^{(i,j)} \mid i \in [d], j \in \{j_1, j_2, \ldots, j_r\}, \boldsymbol{\Phi}_{i'j'}^{(i,j)} = \mathbb{1}[i' \in \{i,j\}, j' \in \{i,j\} \setminus \{i'\}] \right\}$$

This forms a basis to generate any matrix with independent columns along the indexed set. Hence, the span of $\mathcal{S}_{\boldsymbol{\Phi}^*}$ induces a subspace of symmetric matrices of dimension $\frac{r(r+1)}{2}$ in the vector space $\mathsf{symm}(\mathbb{R}^p)$, i.e. the column vectors along the indexed set is spanned by elements of $\mathcal{S}_{\boldsymbol{\Phi}^*}$. Thus, it is clear that picking a feedback set of size $\frac{r(r+1)}{2} - 1$ in the orthogonal complement of $\boldsymbol{\Phi}^*$, i.e. $\mathcal{O}_{\boldsymbol{\Phi}^*}$ restricted by this span sufficiently teaches $\boldsymbol{\Phi}^*$ if $\mathsf{null}(\boldsymbol{\Phi}^*)$ is known. One exact form of this set is proven in Lemma 3. Since any solution $\boldsymbol{\Phi}'$ is agnostic to the scaling of the target matrix $\boldsymbol{\Phi}'$, we have shown that the sufficiency on the feedback complexity for $\boldsymbol{\Phi}^*$ up to feature equivalence.

Now, we show that the stated feedback set size is necessary. The argument is similar to the proof of Lemma 5.

For the sake of contradiction assume that there is a smaller sized feedback set $\mathcal{F}_{\mathsf{small}}$. This implies that there is some matrix in $\mathsf{VS}(\mathcal{M}_{\mathsf{F}}, \mathcal{F}_{\mathsf{null}})$, a subspace induced by span $\mathcal{S}_{\boldsymbol{\Phi}^*}$, orthogonal to $(\boldsymbol{\Phi}^*)$ is not in the span of $\mathcal{F}_{\mathsf{small}}$, denoted as $\boldsymbol{\Phi}'$. If $\boldsymbol{\Phi}'$ is PSD then it is a solution to Eq. (7) and $\boldsymbol{\Phi}'$ is not a scalar multiple of $\boldsymbol{\Phi}^*$. Now, if $\boldsymbol{\Phi}'$ is not PSD we show that there exists scalar $\lambda > 0$ such that

$$\boldsymbol{\Phi}^* + \lambda \boldsymbol{\Phi}' \in \mathsf{Sym}_+(\mathbb{R}^{p \times p}),$$

i.e. the sum is PSD. Consider the eigendecompostion of $\boldsymbol{\Phi}'$ (assume $\mathsf{rank}(\boldsymbol{\Phi}') = r'$)

$$\boldsymbol{\Phi}' = \sum_{i=1}^{r'} \delta_i \mu_i \mu_i^\top$$

for orthogonal eigenvectors $\{\mu_i\}_{i=1}^{r'}$ and the corresponding eigenvalues $\{\delta_i\}_{i=1}^{r'}$. Since (assume) $r_0 \leq r'$ of the eigenvalues are negative we can rewrite $\boldsymbol{\Phi}'$ as

$$\boldsymbol{\Phi}' = \sum_{i=1}^{r_0} \delta_i \mu_i \mu_i^\top + \sum_{j=r_0+1}^{r'} \delta_j \mu_j \mu_j^\top$$

Thus, if we can regulate the values of $\mu_i^\top \boldsymbol{\Phi}^* \mu_i$, for all $i = 1, 2, \ldots, r_0$, noting they are positive, then we can find an appropriate scalar $\lambda > 0$. Let $m^* := \min_{i \in [r_0]} \mu_i^\top \boldsymbol{\Phi}^* \mu_i$ and $\ell^* := \max_{i \in [r_0]} |\delta_i|$. Now, setting $\lambda \leq \frac{m^*}{\ell^*}$ achieves the desired property of $\boldsymbol{\Phi}^* + \lambda \boldsymbol{\Phi}'$ as shown in the proof of Lemma 5.

Consider that both $\boldsymbol{\Phi}'$ and $\boldsymbol{\Phi}^*$ are orthogonal to every element in the feedback set $\mathcal{F}_{\mathsf{small}}$. This orthogonality implies that $\boldsymbol{\Phi}^*$ is not a unique solution to equation Eq. (7) up to a positive scaling factor.

Therefore, we have demonstrated that when the null set $\mathsf{null}(\boldsymbol{\Phi}^*)$ of the target matrix $\boldsymbol{\Phi}^*$ is known, a feedback set of size exactly $\frac{r(r+1)}{2} - 1$ is both necessary and sufficient. $\square$

### F.3. Proof of Lemma 3 and construction of feedback set for $\mathsf{Ker}(\boldsymbol{\Phi}^*)$

Up until this point we haven's shown how to construct this $\frac{r(r+1)}{2} - 1$ sized feedback set. Consider the following union:

$$\left\{ v_1 v_1^\top \right\} \cup \left\{ v_2 v_2^\top, (v_2 + v_1)(v_2 + v_1)^\top \right\} \cup \ldots \cup \left\{ v_r v_r^\top, (v_1 + v_r)(v_1 + v_r)^\top, \ldots, (v_{r-1} + v_r)(v_{r-1} + v_r)^\top \right\}$$

We can show that this union is a set of linearly independent matrices of rank 1 as stated in Lemma 3 below.

**Lemma 3.** *Let $\{v_i\}_{i=1}^r \subset \mathbb{R}^p$ be a set of orthogonal vectors. Then, the set of rank-1 matrices*

$$\mathcal{B} := \left\{ v_i v_i^\top, \ (v_i + v_j)(v_i + v_j)^\top \ \middle| \ 1 \leq i < j \leq r \right\}$$

*is linearly independent in the space of symmetric matrices $\mathsf{Sym}(\mathbb{R}^{p \times p})$.*

*Proof.* We prove the claim by considering two separate cases. For the sake of contradiction, suppose that the set $\mathcal{B}$ is linearly dependent. This implies that there exists at least one matrix of the form $v_i v_i^\top$ or $(v_i + v_j)(v_i + v_j)^\top$ that can be expressed as a linear combination of the other matrices in $\mathcal{B}$. We now examine these two cases individually.

**Case 1**: First, we assume that for some $i \in [r]$, $v_i v_i^\top$ can be written as a linear combination. Thus, there exists scalars that satisfy the following property

$$v_i v_i^\top = \sum_{j=1}^{r'} \alpha_j v_{i_j} v_{i_j}^\top + \sum_{k=1}^{r''} \beta_k (v_{l_k} + v_{m_k})(v_{l_k} + v_{m_k})^\top \tag{8}$$

$$\forall j, k, \quad \alpha_j, \beta_k > 0, i_j \neq i, l_k < m_k \tag{9}$$

Now, note that we can write

$$\sum_{k=1}^{r''} \beta_k (v_{l_k} + v_{m_k})(v_{l_k} + v_{m_k})^\top = \sum_{k=1,l_k=i}^{r''} \beta_k (v_{l_k} + v_{m_k})v_{l_k}^\top + \sum_{k=1,l_k\neq i}^{r''} \beta_k (v_{l_k} + v_{m_k})v_{l_k}^\top + \sum_{k=1}^{r''} \beta_k (v_{l_k} + v_{m_k})v_{m_k}^\top$$

But the following sum

$$\sum_{j=1}^{r'} \alpha_j v_{i_j} v_{i_j}^\top + \sum_{k=1,l_k\neq i}^{r''} \beta_k (v_{l_k} + v_{m_k})v_{l_k}^\top + \sum_{k=1}^{r''} \beta_k (v_{l_k} + v_{m_k})v_{m_k}^\top$$

doesn't span (as column vectors) a subspace that contains the column vector $v_i$ because $\{v_i\}_{i=1}^r$ is a set of orthogonal vectors. Thus, we can write

$$v_i v_i^\top = \sum_{k=1,l_k=i}^{r''} \beta_k (v_{l_k} + v_{m_k})v_{l_k}^\top = \left( \sum_{k=1,l_k=i}^{r''} \beta_k v_{l_k} + \sum_{k=1,l_k=i}^{r''} \beta_k v_{m_k} \right) v_i^\top \tag{10}$$

This implies that

$$\sum_{k=1,l_k=i}^{r''} \beta_k v_{m_k} = 0 \implies \text{if } l_k = i, \beta_k = 0 \tag{11}$$

Since not all $\beta_k = 0$ corresponding to $l_k = i$ (otherwise $\sum_{k=1,l_k=i}^{r''} \beta_k v_{l_k} = 0$) we have shown that $v_i v_i^\top$ can not be written as a linear combination of elements in $\mathcal{B} \setminus \{v_i v_i^\top\}$.

**Case 2**: Now, we consider the second case where there exists some indices $i, j$ such that $(v_i + v_j)(v_i + v_j)^\top$ is a sum of linear combination of elements in $\mathcal{B}$. Note that this linear combination can't have an element of type $v_k v_k^\top$ as it contradicts the first case. So, there are scalars such that

$$(v_i + v_j)(v_i + v_j)^\top = \sum_{k=1}^{r''} \beta_k (v_{l_k} + v_{m_k})(v_{l_k} + v_{m_k})^\top \tag{12}$$

$$\forall k, \quad l_k < m_k \tag{13}$$

But we rewrite this as

$$(v_i + v_j)v_i^\top + (v_i + v_j)v_j^\top$$

$$= \sum_{k=1,l_k=i}^{r''} \beta_k (v_i + v_{m_k})v_i^\top + \sum_{k=1,m_k=j}^{r''} \beta_k (v_{l_k} + v_j)v_j^\top + \sum_{\substack{k=1,l_k\neq i, \\ m_k\neq j}}^{r''} \beta_k (v_{l_k} + v_{m_k})(v_{l_k} + v_{m_k})^\top$$

Note that if $l_k = i$ then the corresponding $m_k \neq j$ and vice versa. Since $\{v_i\}_{i=1}^r$ are orthogonal, the decomposition above implies

$$(v_i + v_j)v_i^\top = \sum_{k=1, l_k=i}^{r''} \beta_k (v_i + v_{m_k}) v_i^\top \tag{14}$$

$$(v_i + v_j)v_j^\top = \sum_{k=1, m_k=j}^{r''} \beta_k (v_{l_k} + v_j) v_j^\top \tag{15}$$

$$\sum_{\substack{k=1, l_k \neq i, \\ m_k \neq j}}^{r''} \beta_k (v_{l_k} + v_{m_k})(v_{l_k} + v_{m_k})^\top = 0 \tag{16}$$

But using the arguments in Eq. (10) and Eq. (11), we can achieve Eq. (14) or Eq. (15).

Thus, we have shown that the set of rank-1 matrices as described in $\mathcal{B}$ are linearly independent. $\qquad\square$

In Lemma 7, we discussed that in order to teach $\boldsymbol{\Phi}^*$ sufficiently agent needs a feedback set of size $\frac{r(r+1)}{2} - 1$ if the null set of $\boldsymbol{\Phi}^*$ is known. We can establish this feedback set using the basis shown in Lemma 3. We state this result in the following lemma.

**Lemma 9.** *For a given target matrix $\boldsymbol{\Phi}^* = \sum_{i=1}^r \gamma_i v_i v_i^\top$ and basis set of matrices $\mathcal{B}$ as shown in Lemma 3, the following set spans a subspace of dimension $\frac{r(r+1)}{2} - 1$ in $\mathsf{Sym}(\mathbb{R}^{p \times p})$.*

$$\mathcal{O}_\mathcal{B} := \left\{ \begin{array}{l} v_1 v_1^\top - \lambda_{11} yy^\top, v_2 v_2^\top - \lambda_{22} yy^\top, (v_1 + v_2)(v_1 + v_2)^\top - \lambda_{12} yy^\top, \ldots, \\ v_r v_r^\top - \lambda_{rr} yy^\top, (v_1 + v_r)(v_1 + v_r)^\top - \lambda_{1r} yy^\top, \ldots, \\ (v_{r-1} + v_r)(v_{r-1} + v_r)^\top - \lambda_{(r-1)r} yy^\top \end{array} \right\}$$

$$y\boldsymbol{\Phi}^* y^\top \neq 0$$

$$\forall i, j, \quad \lambda_{ii} = \frac{v_i \boldsymbol{\Phi}^* v_i^\top}{y \boldsymbol{\Phi}^* y^\top}, \quad \lambda_{ij} = \frac{(v_i + v_j)\boldsymbol{\Phi}^*(v_i + v_j)^\top}{y \boldsymbol{\Phi}^* y^\top} \quad (i \neq j)$$

*Proof.* Since $\boldsymbol{\Phi}^*$ has at least $r$ positive eigenvalues there exists a vector $y \in \mathbb{R}^p$ such that $y\boldsymbol{\Phi}^* y^\top \neq 0$. It is straightforward to note that $\mathcal{O}_\mathcal{B}$ is orthogonal to $\boldsymbol{\Phi}^*$. As $\mathcal{O}_\mathcal{B} \subset \mathrm{span}\langle \mathcal{B} \rangle$ and $\boldsymbol{\Phi}^* \perp \mathcal{O}_\mathcal{B}$, $\dim(\mathrm{span}\langle \mathcal{O}_\mathcal{B} \rangle) = \frac{r(r+1)}{2} - 1$. $\qquad\square$

Now, we will complete the proof of the main result of the appendix here.

*Proof of Theorem 1.* Combining the results from Lemma 6, Lemma 7, and Lemma 9, we conclude that the feedback setup in Eq. (2) can be effectively decomposed into teaching the null space and the span of the eigenvectors of $\boldsymbol{\Phi}^*$. The constructed feedback sets ensure that $\boldsymbol{\Phi}^*$ is uniquely identified up to a linear scaling factor with optimal sample efficiency. $\qquad\square$

# G. Proof of Theorem 1: Lower bound

In this appendix, we provide the proof of the lower bound as stated in Theorem 1. We proceed by first showing some useful properties on a valid feedback set $\mathcal{F}(\mathbb{R}^p, \mathcal{M}_F, \Phi^*)$ for a target feature matrix $\Phi^*$. They are stated in Lemma 10 and Lemma 11.

First, we consider a basic spanning property of matrices $(xx^\top - yy^\top)$ for any pair $(x, y) \in \mathcal{F}$ in the space of symmetric matrices $\mathsf{Sym}(\mathbb{R}^{p \times p})$.

**Lemma 10.** *If* $\Phi \in \mathcal{O}_{\Phi^*}$ *such that span* $\langle \mathsf{col}(\Phi) \rangle \subset span \langle V_{[r]} \rangle$ *then* $\Phi \in span \langle \mathcal{F} \rangle$.

*Proof.* Consider an $\Phi \in \mathcal{O}_{\Phi^*}$ such that span $\langle \mathsf{col}(\Phi) \rangle \subset$ span $\langle V_{[r]} \rangle$. Note that the eigendecompostion of $\Phi$ (assume rank$(\Phi) = r' < r$)

$$\Phi = \sum_{i=1}^{r'} \delta_i \mu_i \mu_i^\top$$

for orthogonal eigenvectors $\{\mu_i\}_{i=1}^{r'}$ and the corresponding eigenvalues $\{\delta_i\}_{i=1}^{r'}$ has the property that $span \left\langle \{\mu_i\}_{i=1}^{r'} \right\rangle \subset$ span $\langle V_{[r]} \rangle$. Using the arguments exactly as shown in the second half of the proof of Lemma 7 we can show there exists $\lambda > 0$ such that $\Phi^* + \lambda \Phi \in \mathsf{VS}(\mathcal{F}, \mathcal{M}_F)$. But then $\Phi$ is not feature equivalent to $\Phi^*$. But this contradicts the assumption of $\mathcal{F}$ being a valid feedback set. $\square$

**Lemma 11.** *There exists vectors* $U_{[p-r]} \subset$ null$(\Phi^*)$ *(of size* $p - r$*) such that span* $\langle U_{[p-r]} \rangle =$ null$(\Phi^*)$ *and for any vector* $v \in U_{[p-r]}$, $vv^\top \in span \langle \mathcal{F} \rangle$.

*Proof.* Assuming the contrary, there exists $v \in$ span $\langle$null$(\Phi^*)\rangle$ such that $vv^\top \notin$ span $\langle \mathcal{F} \rangle$.

Now if $vv^\top \perp \mathcal{F}$, then for any scalar $\lambda > 0$, $\Phi^* + \lambda vv^\top$ is both symmetric and positive semi-definite and satisfies all the conditions in Eq. (1) wrt $\mathcal{F}$ a contradiction as $\Phi^* + \lambda vv^\top$ is not feature equivalent to $\Phi^*$.

So, consider the case when $vv^\top \not\perp \mathcal{F}$. Let $\{v_{r+1}, \dots, v_{p-1}\}$ be an orthogonal extension[3] of $v$ such that $\{v_{r+1}, \dots, v_{p-1}, v\}$ forms a basis of null$(\Phi^*)$, i.e., in other words

$$v \perp \{v_{r+1}, \dots, v_{p-1}\} \quad \& \quad span \langle \{v_{r+1}, \dots, v_{p-1}, v\} \rangle = \text{null}(\Phi^*).$$

We will first show that there exists some $\Phi'$ ($\neq \lambda \Phi^*$, for some $\lambda > 0$) $\in \mathsf{Sym}(\mathbb{R}^{p \times p})$ orthogonal to $\mathcal{F}$ and furthermore $\{v_{r+1}, \dots, v_{p-1}\} \subset$ null$(\Phi')$.

Consider the intersection (in the space $\mathsf{Sym}(\mathbb{R}^{p \times p})$) of the orthogonal complement of the matrices $\{v_{r+1}v_{r+1}^\top, \dots, v_{p-1}v_{p-1}^\top\}$, denote it as $\mathcal{O}_{\mathsf{rest}}$, i.e.,

$$\mathcal{O}_{\mathsf{rest}} := \bigcap_{i=r+1}^{p-1} \mathcal{O}_{v_i v_i^\top}$$

Note that

$$\dim(\mathcal{O}_{\mathsf{rest}}) = p(p+1)/2 - p + r$$

Since $vv^\top$ is in $\mathcal{O}_{\mathsf{rest}}$ and $\dim(\mathcal{O}_{\mathsf{rest}}) > 1$ there exists some $\Phi'$ such that $\Phi' \perp \Phi^*$, and also orthogonal to elements in the feedback set $\mathcal{F}$. Thus, $\Phi'$ has a null set which includes the subset $\{v_{r+1}, \dots, v_{p-1}\}$.

Now, the rest of the proof involves showing existence of some scalar $\lambda > 0$ such that $\Phi^* + \lambda \Phi'$ satisfies the conditions of Eq. (1) for the feedback set $\mathcal{F}$. Note that if $v\Phi'v^\top = 0$ then the proof is straightforward as span $\langle \{v_{r+1}, \dots, v_{p-1}, v\} \rangle \subset$ null$(\Phi')$, which implies span $\langle \mathsf{col}(\Phi') \rangle \subset$ span $\langle V_{[r]} \rangle$. But this is precisely the condition for Lemma 10 to hold.

---

[3]the set is not trivially empty in which case the proof follows easily

Without loss of generality assume that $v\boldsymbol{\Phi}'v^\top > 0$. First note that the eigendecomposition of $\boldsymbol{\Phi}'$ has eigenvectors that are contained in $V_{[r]} \cup \{v\}$. Consider some arbitrary choice of $\lambda > 0$, we will fix a value later. It is straightforward that $\boldsymbol{\Phi}^* + \lambda\boldsymbol{\Phi}'$ is symmetric for $\boldsymbol{\Phi}^*$ and $\boldsymbol{\Phi}'$ are symmetric. In order to show it is positive semi-definite, it suffices to show that

$$\forall u \in \mathbb{R}^p, u^\top(\boldsymbol{\Phi}^* + \lambda\boldsymbol{\Phi}')u \geq 0 \tag{17}$$

Since $\{v_{r+1}, \ldots, v_{p-1}\} \subset (\text{null}(\boldsymbol{\Phi}^*) \cap \text{null}(\boldsymbol{\Phi}'))$ we can simplify Eq. (17) to

$$\forall u \in \text{span}\left\langle V_{[r]} \cup \{v\}\right\rangle, u^\top(\boldsymbol{\Phi}^* + \lambda\boldsymbol{\Phi}')u \geq 0 \tag{18}$$

Consider the decomposition of any arbitrary vector $u \in \text{span}\left\langle V_{[r]} \cup \{v\}\right\rangle$ as follows:

$$u = u_{[r]} + v', \text{ such that } u_{[r]} \in \text{span}\left\langle V_{[r]}\right\rangle, v' \in \text{span}\left\langle\{v\}\right\rangle \tag{19}$$

$$u_{[r]} := \sum_{i=1}^{r} \alpha_i v_i, \quad \forall i \ \alpha_i \in \mathbb{R} \tag{20}$$

From here on we assume that $u_{[r]} \neq 0$. The alternate case is trivial as $v'^\top\boldsymbol{\Phi}'v' > 0$.

Now, we write the vectors as scalar multiples of their corresponding unit vectors

$$u_{[r]} = \delta_r \cdot \hat{u}_r, \quad \hat{u}_r := \frac{u_{[r]}}{||u_{[r]}||^2_{V_{[r]}}}, ||u_{[r]}||^2_{V_{[r]}} := \sum_{i=1}^{r} \alpha_i^2 \tag{21}$$

$$v' = \delta_{v'} \cdot \hat{v}, \quad \hat{v} := \frac{v}{||v||_2^2} \tag{22}$$

*Remark*: Although we have computed the norm of $u_{[r]}$ as $||u_{[r]}||^2_{V_{[r]}}$ in the orthonormal basis $V_{[r]}$, note that the norm remains unchanged (same as the $\ell_2$). $\ell_2$ is used for ease of analysis later on.

Using the decomposition in Eq. (19)-(20), we can write Eq. (18) as

$$\begin{aligned} u^\top(\boldsymbol{\Phi}^* + \lambda\boldsymbol{\Phi}')u &= (u_{[r]} + v')^\top(\boldsymbol{\Phi}^* + \lambda\boldsymbol{\Phi}')(u_{[r]} + v') \\ &= u_{[r]}^\top\boldsymbol{\Phi}^*u_{[r]} + \lambda(u_{[r]} + v')^\top\boldsymbol{\Phi}'(u_{[r]} + v') \\ &= \delta_r^2 \cdot \hat{u}_r^\top\boldsymbol{\Phi}^*\hat{u}_r + \lambda\big(\delta_r^2 \cdot \hat{u}_r^\top\boldsymbol{\Phi}'\hat{u}_r + 2\delta_r\delta_{v'} \cdot \hat{u}_r^\top\boldsymbol{\Phi}'\hat{v} + \delta_{v'}^2 \cdot \hat{v}^\top\boldsymbol{\Phi}'\hat{v}\big) \end{aligned} \tag{23}$$

Since we want $u^\top(\boldsymbol{\Phi}^* + \lambda\boldsymbol{\Phi}')u \geq 0$ we can further simplify Eq. (23) as

$$\hat{u}_r^\top\boldsymbol{\Phi}^*\hat{u}_r + \lambda\left(\hat{u}_r^\top\boldsymbol{\Phi}'\hat{u}_r + 2\frac{\delta_r\delta_{v'}}{\delta_r^2} \cdot \hat{u}_r^\top\boldsymbol{\Phi}'\hat{v} + \frac{\delta_{v'}^2}{\delta_r^2} \cdot \hat{v}^\top\boldsymbol{\Phi}'\hat{v}\right) \underset{?}{\geq} 0 \tag{24}$$

$$\iff \underbrace{\hat{u}_r^\top\boldsymbol{\Phi}^*\hat{u}_r}_{\color{red}(1)} + \lambda\left(\underbrace{\hat{u}_r^\top\boldsymbol{\Phi}'\hat{u}_r}_{\color{blue}(3)} + \underbrace{2\xi \cdot \hat{u}_r^\top\boldsymbol{\Phi}'\hat{v} + \xi^2 \cdot \hat{v}^\top\boldsymbol{\Phi}'\hat{v}}_{\color{blue}(2)}\right) \underset{?}{\geq} 0 \tag{25}$$

where we have used $\xi = \frac{\delta_{v'}}{\delta_r}$. The next part of the proof we show that ${\color{red}(1)}$ is lower bounded by a positive constant whereas ${\color{blue}(2)}$ is upper bounded by a positive constant and there is a choice of $\lambda$ so that ${\color{blue}(3)}$ is always smaller than ${\color{red}(1)}$.

Considering ${\color{red}(1)}$ we note that $\hat{u}_r$ is a unit vector wrt the orthonormal set of basis $V_{[r]}$. Expanding using the eigendecomposition of Eq. (5)

$$\hat{u}_r^\top\boldsymbol{\Phi}^*\hat{u}_r = \sum_{i=1}^{r} \frac{\alpha_i^2}{\sum_{i=1}^{r}\alpha_i^2} \cdot \gamma_i \geq \min_i \gamma_i > 0$$

The last inequality follows as all the eigenvalues in the eigendecompostion are (strictly) positive. Denote this minimum eigenvalue as $\gamma_{\min} := \min_i \gamma_i$.

Considering ${\color{blue}(2)}$ note that only terms that are variable (i.e. could change value) is $\xi$ as $\hat{u}_r^\top\boldsymbol{\Phi}'\hat{v}$ is

Note that $\hat{v}$ is a fixed vector and $\hat{u}_r$ has a fixed norm (using Eq. (21)-(22)), so $|\hat{u}_r^\top \mathbf{\Phi}' \hat{v}| \leq C$ for some bounded constant $C > 0$ whereas $\hat{v}^\top \mathbf{\Phi}' \hat{v}$ is already a constant. Now, $|2\xi \cdot \hat{u}_r^\top \mathbf{\Phi}' \hat{v}|$ exceeds $\xi^2 \cdot \hat{v}^\top \mathbf{\Phi}' \hat{v}$ only if

$$|2\xi \cdot \hat{u}_r^\top \mathbf{\Phi}' \hat{v}| \geq |\xi^2 \cdot \hat{v}^\top \mathbf{\Phi}' \hat{v}| \iff \frac{|\hat{u}_r^\top \mathbf{\Phi}' \hat{v}|}{\hat{v}^\top \mathbf{\Phi}' \hat{v}} \geq \xi \implies \frac{C}{\hat{v}^\top \mathbf{\Phi}' \hat{v}} \geq \xi$$

Rightmost inequality implies that $2\xi \cdot \hat{u}_r^\top \mathbf{\Phi}' \hat{v} + \xi^2 \cdot \hat{v}^\top \mathbf{\Phi}' \hat{v}$ is negative only for an $\xi$ bounded from above by a positive constant. But since $\xi$ is non-negative

$$|2\xi \cdot \hat{u}_r^\top \mathbf{\Phi}' \hat{v} + \xi^2 \cdot \hat{v}^\top \mathbf{\Phi}' \hat{v}| \leq C'(\text{bounded constant})$$

Now using an argument similar to the second half of the proof of Lemma 7, it is straight forward to show that there is a choice of $\lambda' > 0$ so that (3) is always smaller than (1).

Now, for $\lambda = \frac{\lambda'}{2\lceil C' \rceil \lambda''}$ where $\lambda''$ is chosen so that $\lambda_{\min} \geq \frac{\lambda'}{\lambda''}$, we note that

$$\hat{u}_r^\top \mathbf{\Phi}^* \hat{u}_r + \lambda \left( \hat{u}_r^\top \mathbf{\Phi}' \hat{u}_r + 2\xi \cdot \hat{u}_r^\top \mathbf{\Phi}' \hat{v} + \xi^2 \cdot \hat{v}^\top \mathbf{\Phi}' \hat{v} \right) \geq \lambda_{\min} + \frac{\lambda'}{2\lceil C' \rceil \lambda''} \hat{u}_r^\top \mathbf{\Phi}' \hat{u}_r - \frac{\lambda'}{2\lambda''} > 0.$$

Using the equivalence in Eq. (23), Eq. (24) and Eq. (25), we have a choice of $\lambda > 0$ such that $u^\top (\mathbf{\Phi}^* + \lambda \mathbf{\Phi}')u \geq 0$ for any arbitrary vector $u \in \text{span} \langle V_{[r]} \cup \{v\} \rangle$. Hence, we have achieved the conditions in Eq. (18), which is the simplification of Eq. (17). This implies that $\mathbf{\Phi}^* + \lambda \mathbf{\Phi}'$ is positive semi-definite.

This implies that there doesn't exist a $v \in \text{span} \langle \text{null}(\mathbf{\Phi}^*) \rangle$ such that $vv^\top \notin \text{span} \langle \mathcal{F} \rangle$ otherwise the assumption on $\mathcal{F}$ to be an oblivious feedback set for $\mathbf{\Phi}^*$ is violated. Thus, the statement of Lemma 11 has to hold. $\qquad \square$

### G.1. Proof of lower bound in Theorem 1

In the following, we provide proof of the main statement on the lower bound of the size of a feedback set.

If any of the two lemmas (10-11) are violated, we can show there exists $\lambda > 0$ and $\mathbf{\Phi}$ such that $\mathbf{\Phi}^* + \lambda \mathbf{\Phi} \in \text{VS}(\mathcal{F}, \mathcal{M}_\mathsf{F})$. In order to ensure these statements, the feedback set should have $\left( \frac{r(r+1)}{2} + (d-r) - 1 \right)$ many elements which proves the lower bound on $\mathcal{F}$.

But using Lemma 7 and Lemma 9 we know that the dimension of the span of matrices that satisfy the condition in Lemma 10 is at the least $\frac{r(r+1)}{2} - 1$. We can use Lemma 9 where $y = \sum_{i=1}^r v_r$ (note $\mathbf{\Phi}^* v \neq 0$). Thus, any basis matrix in $\mathcal{O}_\mathcal{B}$ satisfy the conditions in Lemma 10.

Since the dimension of $\text{null}(\mathbf{\Phi}^*)$ is at least $(d-r)$ thus there are at least $(d-r)$ directions or linearly independent matrices (in $\text{Sym}(\mathbb{R}^{p \times p})$) that need to be spanned by $\mathcal{F}$.

Thus, Lemma 10 implies there are $\frac{r(r+1)}{2} - 1$ linearly independent matrices (in $\mathcal{O}_{\mathbf{\Phi}^*}$) that need to be spanned by $\mathcal{F}$. Similarly, Lemma 11 implies there are $p - r$ linearly independent matrices (in $\mathcal{O}_{\mathbf{\Phi}^*}$) that need to be spanned by $\mathcal{F}$. Note that the column vectors of these matrices from the two statements are spanned by orthogonal set of vectors, i.e. one by $V_{[r]}$ and the other by $\text{null}(\mathbf{\Phi}^*)$ respectively. Thus, these $\frac{r(r+1)}{2} - 1 + (p - r)$ are linearly independent in $\text{Sym}(\mathbb{R}^{p \times p})$, but this forces a lower bound on the size of $\mathcal{F}$ (a lower dimensional span can't contain a set of vectors spanning higher dimensional space). This completes the proof of the lower bound in Theorem 1.

## H. Proof of Theorem 3: General Activations Sampling

We aim to establish both upper and lower bounds on the feedback complexity for oblivious learning in Algorithm 2. The proof revolves around the linear independence of certain symmetric matrices derived from random representations and the dimensionality required to span a target feature matrix.

Let us define a positive index $P = \frac{p(p+1)}{2}$. The agent receives $P$ representations:

$$\mathcal{V}_n := \{v_1, v_2, \ldots, v_P\} \sim \mathcal{D}_\mathcal{V}.$$

For each $i$, we define the symmetric matrix $V_i = v_i v_i^\top$.

Consider the matrix $\mathbb{M}$ formed by concatenating the vectorized $V_i$:

$$\mathbb{M} = \begin{bmatrix} \text{vec}(V_1) & \text{vec}(V_2) & \cdots & \text{vec}(V_P) \end{bmatrix},$$

where each $\text{vec}(V_i)$ is treated as a column vector in $\mathbb{R}^P$. The vectorization operation for a symmetric matrix $A \in \mathsf{Sym}(\mathbb{R}^{p \times p})$ is defined as:

$$\text{vec}(A)_k = \begin{cases} A_{ii} & \text{if } k \text{ corresponds to } (i,i), \\ A_{ij} + A_{ji} & \text{if } k \text{ corresponds to } (i,j), \ i < j. \end{cases}$$

The determinant $\det(\mathbb{M})$ is a non-zero polynomial in the entries of $v_1, v_2, \ldots, v_P$. Since the vectors $v_i$ are drawn from a continuous distribution $\mathcal{D}_\mathcal{V}$, using Sard's theorem the probability that $\det(\mathbb{M}) = 0$ is zero, i.e.,

$$\mathcal{P}_{\mathcal{V}_n}(\det(\mathbb{M}) = 0) = 0.$$

This implies that, with probability 1, the set $\{V_1, V_2, \ldots, V_P\}$ is linearly independent in $\mathsf{Sym}(\mathbb{R}^{p \times p})$:

$$\mathcal{P}_{\mathcal{V}_n}\left(\{v_i v_i^\top\} \text{ is linearly independent in } \mathsf{Sym}(\mathbb{R}^{p \times p})\right) = 1. \tag{26}$$

Next, let $\Sigma^* \neq 0$ be an arbitrary target feature matrix for learning with feedback in Algorithm 2. Without loss of generality, assume $v := v_1 \neq 0$. Define the set $\mathcal{F}$ of rescaled pairs as:

$$\mathcal{F} = \left\{ (v, \sqrt{\gamma_i} v_i) \ \middle| \ \Sigma^* \cdot \left(vv^\top - \gamma_i v_i v_i^\top\right) = 0, \ \sqrt{\gamma_i} > 0 \right\},$$

noting that $|\mathcal{F}| = P - 1$.

Assume, for contradiction, that the elements of $\mathcal{F}$ are linearly dependent in $\mathsf{Sym}(\mathbb{R}^{p \times p})$. Then, there exist scalars $\{a_i\}$ (not all zero) such that:

$$\sum_{i=2}^{P} a_i \left(vv^\top - \gamma_i v_i v_i^\top\right) = 0 \quad \Rightarrow \quad \left(\sum_{i=2}^{P} a_i\right) vv^\top = \sum_{i=2}^{P} a_i \gamma_i v_i v_i^\top.$$

However, since $\{v_i v_i^\top\}$ are linearly independent with probability 1, it must be that:

$$\sum_{i=2}^{P} a_i = 0 \quad \text{and} \quad a_i \gamma_i = 0 \quad \forall i.$$

Given that $\gamma_i > 0$, this implies $a_i = 0$ for all $i$, contradicting the assumption of linear dependence. Therefore, matrices induced by $\mathcal{F}$ are linearly independent.

This implies that $\mathcal{F}$ induces a set of linearly independent matrices, i.e., $\{vv^\top - \gamma_i v_i v_i^\top\}$ in the orthogonal complement $\mathcal{O}_{\Sigma^*}$, and since $\Sigma^*$ has at most $P$ degrees of freedom, any matrix $\Sigma' \in \mathsf{Sym}(\mathbb{R}^{p \times p})$ satisfying:

$$\Sigma' \cdot \left(vv^\top - \gamma_i v_i v_i^\top\right) = 0 \quad \forall i$$

must be a positive scalar multiple of $\Sigma^*$.

Thus, using Eq. (26), with probability 1, the feedback set $\mathcal{F}$ is valid:

$$\mathcal{P}_{\mathcal{V}_n} \left( \mathcal{F} \text{ is a valid feedback set} \right) = 1.$$

Since $\Sigma^*$ was arbitrary, the worst-case feedback complexity is almost surely upper bounded by $P - 1$ for achieving feature equivalence.

For the lower bound, consider the proof of the lower bound in Theorem 1, specifically Lemma 10, which asserts that for any feedback set $\mathcal{F}$ in Algorithm 1, given any target matrix $\Sigma^* \in \mathsf{Sym}(\mathbb{R}^{p \times p})$, if $\Sigma \in \mathcal{O}_{\Sigma^*}$ such that span $\langle \mathsf{col}(\Sigma) \rangle \subset$ span $\langle Z_{[r]} \rangle$ then $\Sigma \in$ span $\langle \mathcal{F} \rangle$ where $Z_{[r]}$ $(r \leq d)$ is defined as the set of eigenvectors in the eigendecompostion of $\Sigma^*$ (see Eq. (5)).

This implies that any feedback set $\mathcal{F}(\mathcal{V}_n, \Sigma^*)$ must span certain matrices $\Sigma' \in \mathsf{Sym}(\mathbb{R}^{p \times p})$. Suppose the agent receives $\ell$ representations $v_1, v_2, \ldots, v_\ell \sim \mathcal{D}_{\mathcal{V}}$ and constructs:

$$\mathbb{M} = \begin{bmatrix} \mathrm{vec}(\Sigma') & \mathrm{vec}(V_1) & \cdots & \mathrm{vec}(V_\ell) \end{bmatrix}.$$

Now, consider the polynomial equation $\det(\mathbb{M}) = 0$. Since every entry of $\mathbb{M}$ is semantically different, the determinant $\det(\mathbb{M})$ is a non-zero polynomial. Note that there are $\frac{p(p+1)}{2}$ many degrees of freedom for the rows. Thus, it is clear that the zero set $\{\det(\mathbb{M}) = 0\}$ has Lebesgue measure zero if $\ell < \frac{p(p+1)}{2}$, i.e. $\mathbb{M}$ requires at least $\frac{p(p+1)}{2}$ columns for $\det(\mathbb{M})$ to be identically zero. But this implies that set $\{v_i v_i^\top\}_{i=1}^\ell$ can't span $\Sigma'$ (almost surely) if $\ell \leq \frac{p(p+1)}{2} - 1$. Hence, (almost surely) the agent can't devise a feedback set for oblivious learning in Algorithm 2. In other words, if $\ell \leq \frac{p(p+1)}{2} - 1$,

$$\mathcal{P}_{\mathcal{V}_\ell} \left( \text{agent devises a feedback set } \mathcal{F} \text{ up to feature equivalence} \right) = 0$$

Hence, to span $\Sigma'$, it almost surely requires at least $\frac{p(p+1)}{2}$ representations. Therefore, the feedback complexity cannot be lower than $\Omega\left(\frac{p(p+1)}{2}\right)$.

Combining the upper and lower bounds, we conclude that the feedback complexity for oblivious learning in Algorithm 2 is tightly bounded by $\Theta\left(\frac{p(p+1)}{2}\right)$.

# I. Proof of Theorem 4: Sparse Activations Sampling

Here we consider the analysis for the case when the activations $\mathcal{V}$ are sampled from the sparse distribution as stated in Definition 1.

In Theorem 4, we assume that the activations are sampled from a Lebesgue distribution. This, sufficiently, ensures that (almost surely) any random sampling of $P$ activations induces a set of linearly independent rank-1 matrices. Since the distribution in Assumption 1 is not a Lebesgue distribution over the entire support $[0, 1]$, requiring an understanding of certain events of the sampling of activations which could lead to linearly independent rank-1 matrices.

In the proof of Theorem 2, we used a set of sparse activations using the standard basis of the vector space $\mathbb{R}^p$. We note that the idea could be generalized to arbitrary choice of scalars as well, i.e.,

$$U_g = \{\lambda_i e_i : \lambda_i \neq 0, 1 \leq i \leq p\} \cup \{(\lambda_{iji} e_i + \lambda_{ijj} e_j) : \lambda_{iji}, \lambda_{ijj} \neq 0, 1 \leq i < j \leq p\}.$$

Here $e_i$ is the $i$th standard basis vector. Note that the corresponding set of rank-1 matrices, denoted as $\widehat{U}_g$

$$\widehat{U}_g = \left\{ \lambda_i^2 e_i e_i^T : 1 \leq i \leq p \right\} \cup \left\{ (\lambda_{iji} e_i + \lambda_{ijj} e_j)(\lambda_{iji} e_i + \lambda_{ijj} e_j)^T : 1 \leq i < j \leq p \right\}$$

is linearly independent in the space of symmetric matrices on $\mathbb{R}^p$, i.e., $\mathsf{Sym}(\mathbb{R}^{p \times p})$.

Assume that activations are sampled $P$ times, denoted as $\mathcal{V}_P$. Now, consider the design matrix $\mathbb{M} = \begin{bmatrix} V_1 & V_2 & \ldots V_P \end{bmatrix}$ as shown in the proof of Theorem 3. We know that if $\det(\mathbb{M})$ is non-zero then $\{V_i\}'s$ are linearly independent in $\mathsf{Sym}(\mathbb{R}^{p \times p})$. To show if a sampled set $\mathcal{V}_P$ exhibits this property we need to show that $\det(\mathbb{M})$ is not identically zero, which could be possible for activations sampled from sparse distributions as stated in Assumption 1, i.e. $\mathcal{P}_{v \sim \mathcal{D}_{\mathsf{sparse}}}(v_i \neq 0) > 0$.

Note that $\det(\mathbb{M}) = \sum_{\sigma \in \mathsf{P}_P} \prod_i \mathbb{M}_{i\sigma(i)}$. Consider the diagonal of $\mathbb{M}$. Consider the situation where all the entries are non-zero. This corresponds to sampling a set of activations of the form $\widehat{U}_g$. Consider the following random design matrix $\mathbb{M}$.

$$
\mathbb{M} = \begin{bmatrix}
\lambda_1^2 & \cdot & \cdot & \cdots & \cdot & \lambda_{121}^2 & \cdots & & \cdot \\
\cdot & \lambda_2^2 & \cdot & \cdots & \cdot & \lambda_{122}^2 & \cdots & & \vdots \\
\cdot & \cdot & \lambda_3^2 & \cdots & \cdot & \cdot & \cdots & \lambda_{(p-1)p(p-1)}^2 & \\
\vdots & \cdots & \cdots & \cdots & \lambda_p^2 & \cdots & \cdots & \lambda_{(p-1)pp}^2 & \\
\vdots & \cdots & \cdots & \cdots & \cdot & \lambda_{121}\lambda_{122} & \cdots & & \cdots \\
\vdots & \cdots & \cdots & \cdots & \cdot & \cdot & \cdots & & \cdots \\
\cdot & \cdots & \cdots & \cdots & \cdot & \cdot & \cdot & \lambda_{(p-1)p(p-1)}\lambda_{(p-1)pp}
\end{bmatrix} .
$$

Now, a random design matrix $\mathbb{M}$ is not identically zero for any set of $P$ randomly sampled activations that satisfy the following indexing property:

$$
\mathcal{R} := \{v : v_i \neq 0, 1 \leq i \leq p\} \cup \{v : v_i, v_j \neq 0, 1 \leq i < j \leq p\}. \tag{27}
$$

This is so because for identity permutation, we have $\prod_i \mathbb{M}_{ii} \neq 0$. Now, we will compute the probability that $\mathcal{R}$ is sampled from $\mathcal{D}_{\mathsf{sparse}}$. Using the independence of sampling of each index of an activation, the probabilities for the two subsets of $\mathcal{R}$ can be computed as follows:

- $p$ activations $\{\alpha_1, \alpha_2, \cdots, \alpha_p\} \sim \mathcal{D}_{\mathsf{sparse}}^p$ such that $\alpha_{ii} \neq 0$. Using independence, we have

$$
\mathcal{P}_1 = \sum_{i=0}^{s-1} \binom{p-1}{i} p_{nz}^{i+1} (1 - p_{nz})^{p-1-i},
$$

- Rest of $p(p-1)/2$ activations of $\mathcal{R}$ in Eq. (27) require at least two indices to be non-zero. This could be computed as

$$
\mathcal{P}_2 = \sum_{i=0}^{s-2} \binom{p-2}{i} p_{nz}^{i+2} (1 - p_{nz})^{p-2-i}.
$$

Now, note that these $P$ activations can be permuted in $P!$ ways and thus

$$
\mathcal{P}_{\mathcal{V}_p}(\mathcal{V}_P \equiv \mathcal{R}) \geq P! \cdot \mathcal{P}_1^p \cdot \mathcal{P}_2^{(P-p)} = P! \cdot \underbrace{\left(\sum_{i=0}^{s-1} \binom{p-1}{i} p_{nz}^{i+1} (1-p_{nz})^{p-1-i}\right)^p \left(\sum_{i=0}^{s-2} \binom{p-2}{i} p_{nz}^{i+2} (1-p_{nz})^{p-2-i}\right)^{(P-p)}}_{p_{\mathsf{s}}}
$$

$$
\tag{28}
$$

Now, we will complete the proof of the theorem using Hoeffding's inequality. Assume that the agent samples $N$ activations, we will compute the probability that $\mathcal{R} \subset \mathcal{V}_N$. Consider all possible $P$-subsets of $N$ items, enumerated as $\left\{1, 2, \ldots, \binom{N}{P}\right\}$. Now, define random variables $X_i$ as

$$
X_i = \begin{cases} 1 & \text{if } i\text{th subset equals } \mathcal{R}, \\ 0 & \text{o.w.} \end{cases}
$$

Now, define sum random variable $X = \sum_i^{\binom{N}{P}} X_i$. We want to understand the probability $\mathcal{P}_{\mathcal{V}_N}(X \geq 1)$. Now note that,

$$
\mathbb{E}_{\mathcal{V}_N \sim \mathcal{D}_{\mathsf{sparse}}}[X] = \sum_i \mathbb{E}[X_i] = \binom{N}{P} \cdot \mathcal{P}_{\mathcal{V}_P}(\mathcal{V}_P \equiv \mathcal{R})
$$

Now, using Hoeffding's inequality

$$
\mathcal{P}_{\mathcal{V}_N}(X > 0) \geq 1 - 2\exp^{-2\mathbb{E}[X]^2} \geq 1 - 2\exp^{-2\binom{N}{P}^2 p_{\mathsf{s}}^2}
$$

Now, for a given choice of of $\delta > 0$, we want $\delta \geq 2 \exp^{-2 \binom{N}{P}^2 p_{\mathsf{s}}^2}$. Using Sterling's approximation

$$\binom{N}{P} \geq \frac{1}{p_{\mathsf{s}}} \sqrt{\log \frac{4}{\delta^2}} \implies \left( \frac{eN}{P} \right)^P \geq \frac{1}{p_{\mathsf{s}}} \sqrt{\log \frac{4}{\delta^2}} \implies N \geq \frac{P}{e} \left( \frac{1}{p_{\mathsf{s}}^2} \log \frac{4}{\delta^2} \right)^{1/2P}$$

