# OpenReview forum: "The Complexity of Learning Sparse Superposed Features with Feedback"
_ICML.cc/2025/Conference — ICML 2025 poster_

### Official Review · Reviewer_8xsq · 2025-03-08

**Overall Recommendation:** 4

**Summary:**

In this work, the authors study how well the feature learned by a neural network can be retrieved by the means of some agents, e.g., an LLM, in the form of *relative triplet comparisons*. Formally, leveraging the linear feature decomposition that encodes features in a dictionary, the authors investigate how well a learner provided with sparse triplets given by an agent can identify the relevant features (up to normal transformation). After introducing the protocol with which a learner receives triplets from an agent, the authors provide bounds on the feedback efficiency, i.e., the minimal number of interactions to learn feature matrices (normal transformations of feature dictionary). The authors provide tight bounds in sparse and standard settings with an agent that gives either constructive or distributional sparse triplets.
Finally, the authors provide empirical validation of their results on synthetic and large-scale models.

**Claims And Evidence:**

The theoretical claims are well supported by clear and detailed proofs, and the authors also provide experimental validation of their theory on synthetic tasks and large-scale models.

**Essential References Not Discussed:**

To the best of my knowledge, there are no essential references not discussed in the current submission.

**Experimental Designs Or Analyses:**

The authors provided the code to reproduce their results. A quick inspection of the code seems to indicate that the experiments were well-conducted, although I did not re-run the experiments. The experimental setup is well introduced however, I have some issues with understanding how well the experiments confirm the theoretical results. The current submission would be improved by clearly stating in more detail what empirical confirmation can be seen from the figure. A comparison between the theoretical bound and the one observed in practice could also be of interest.

**Methods And Evaluation Criteria:**

The main contributions of this work are theoretical, and the authors provided detailed proofs as well as experimental computation to validate the theory both in synthetic and large-scale settings. I believe the methods and evaluation criteria make sense for the problem at hand.

**Other Comments Or Suggestions:**

I list below some potential typos:
- Table 1 / first cell: "Stardard Constructive" --> "Standard Constructive"

**Other Strengths And Weaknesses:**

**Strengths**

- The proofs are clear and well-detailed, and notations are defined early for the reader
- The problem considered seems original and is well explained
- I find the proofs techniques elegant with the null space analysis and cardinal computations. I find it refreshing to see the use of algebra.

**Weaknesses**

I list below what I think are weaknesses, but I would be happy to be corrected if I misunderstood some important aspects of the authors' contributions.

- In my opinion, the main weakness of the current work is the contextualization of the theoretical results. More discussion regarding their impact and comparison to the existing literature could greatly improve the current submission (e.g., benefits of the bounds for the feature learning community, numerical impact on the computational resources needed to retrieve neural network features, etc.).

- Moreover, it is not clear to me how the experiments validate the theoretical findings, and since there is no extended intuition/explanation of them, it is quite hard to grasp their impact, and it makes the global work lack some coherence. An idea to improve the submission would be to add some comparison between theoretical vs. empirical bounds.

- A conclusion/discussion section is missing to summarize the contributions and discuss limitations and future work.

Overall, I find the paper well written, and the framework studied is interesting and quite original. I value the technical contributions for the bounds, however, I have trouble seeing the benefits and practical takeaways of the approach in the context of SAE and feature learning because of the lack of explanations/intuition. This is the reason for my current score. I remain open to modifying my score, provided the authors clarify the points mentioned in the weaknesses section.

**Update after rebuttal**: increased score from 3 to 4.

**Questions For Authors:**

Related to weaknesses:

1) How do the bounds translate in terms of computational resources needed to retrieve features learned by neural networks in practice? The theorems indicate the convergence rate, but it could be interesting to compare it to the naive baseline or common methods used in practice. Could the authors elaborate on that?

2) Could the authors add more discussion on the takeaways of theoretical results and empirical validation?

3) Could the authors compare the theoretical and empirical bounds to better understand how well the theory predicts the practice (for instance, on some simple synthetic task)?

4) Could the authors add a conclusion/discussion section at the end of the paper to summarize the contributions and discuss the limitations and room for future work?

**Relation To Broader Scientific Literature:**

I find that related work and prior works are well introduced and compared. The submission's contributions are part of a growing interest to better understand the internal mechanism at play in neural networks, especially in LLMs (e.g., prior works on Sparse Autoencoders, mechanistic interpretability). The protocol to retrieve features learned by a neural network using an agent such as a human or an LLM, along with the bounds on the feedback complexity, seems novel.

**Theoretical Claims:**

The proofs are well thought, detailed, and clear. I find them elegant, and in my opinion, these are the main strengths of this work.
One does not need to be an expert in the research field of the current submission to understand the proofs and the theoretical results.

---

> ### Author Rebuttal · Authors · 2025-03-30
>
> We thank the reviewer for their thoughtful questions and suggestions. Below, we provide detailed responses to the main concerns.
>
> ---
>
> **1. Can the authors compare theoretical vs empirical bounds?**
>
> We perform experiments based on RFM (cf Section 6, p.8) on a monomial regression task to establish a high correlation of theoretical bounds to empirical bounds. A pdf showing plots and a single Jupyter Notebook can be found here: https://anonymous.4open.science/r/rebuttal_ICML_featurelearning-2E0B/
>
> ---
>
> **2. Conclusion/discussion section missing.**
>
> We will complete a discussion section in the main text to summarize the contributions and discuss the limitations and room for future work.
>
> ---
>
> **3. Benefit to the feature learning community and numerical impact on computational resources for feature retrieval.**
>
> First, we compare some known empirical methods for feature retrieval:
>
> - **Numerical Impact:**
>
>   - **SAEs:** SAEs are widely used to recover interpretable feature dictionaries. However, training SAEs is computationally expensive and typically scales as $\mathcal{O}(Tnpd)$, where $T$: number of iterations, $n$: number of samples, $p$: activation dimension, and $d$: input space dimension. [1]
>     E.g.:
>     > "We take a one-layer transformer with a 512-neuron MLP layer, and decompose the MLP activations into relatively interpretable features by training sparse autoencoders on MLP activations from 8 billion data points..." [2]
>
>   - **CRAFT:** CRAFT uses non-negative matrix factorization (NMF) to extract concepts from hidden activations. It solves a dense matrix factorization problem via ADMM with per-iteration complexity $\mathcal{O}(npr)$, where $r$ is the number of latent components [4]. This method requires access to full activations.
>
>   - **Probing:** [5] studies how semantic properties are linearly decodable from hidden representations. While not aimed at full feature recovery but rather interpretable linear concepts, these methods include:
>     - **Mass-Mean:** $\mathcal{O}(np^2 + p^3)$
>     - **Logistic Regression:** $\mathcal{O}(Tnp)$
>     - **Contrast-Consistent Search:** $\mathcal{O}(mp^2)$ ($m$ is the number of contrast pairs)
>
>    - **Our Method:** Compared to the empirical performance of these baselines, our feedback-based method achieves these provable bounds:
>       - **Random/Sparse Sampling:** $\mathcal{O}(p^2)$ or $\tilde{\mathcal{O}}(p^2)$ ($\tilde{\mathcal{O}}$ hides log and sparsity-related factors)
>       - **Eigendecomposition:** $\mathcal{O}(r^2 + p)$ ($r$: the rank of the feature matrix)
>
>
> - **Relevance to Feature Learning:**
>
>   - **Radhakrishnan et al. (2024):** [3] posits a Neural Feature Ansatz, showing a strong correlation between weight outer products and a neural network's Average Gradient Outer Product (AGOP). Our method complements this line of work by providing insights into how efficient feature learning could be possible, leveraging low-dimensional structure of task-specific relevant directions (in AGOP).
>
>   - **Distillation:** Our results are applicable to model distillation, suggesting that more efficient algorithms can be designed by exploiting the low-dimensional structure of the features of larger models.
>
>   - **Feature Formation:** Our work highlights that learning certain classes of features can have inherent statistical and computational bottlenecks, especially in high dimensions. Even with feedback, quadratic dependence on dimension is generally unavoidable.
>
> ---
>
> **4. Takeaways of the theoretical results and empirical validation.**
>
> Key takeaways:
>
> - Our results suggest potential sample complexity lower bounds on feature formation in the activation of a layer or a trained SAE for standard learning frameworks, e.g., active learning, iid learning, machine teaching.
> - There is a clear trade-off between the expressiveness of high-dimensional feature matrices and the difficulty of recovering them: more complex or expressive dictionaries generally require more feedback or data.
> - In the general case, the required feedback scales quadratically with dimension, though this can be reduced under structural assumptions (e.g., low rank).
> - Empirically, we find recovery becomes harder in high dimensions, and results suggest leveraging structure may improve efficiency, motivating future works to dimensionality reduction.
>
> ---
> ### **References**
>
> [1] [Open Problems in Mechanistic Interpretability](https://arxiv.org/abs/2501.16496)
> [2] [Towards Monosemanticity: Dictionary Learning](https://transformer-circuits.pub/2023/monosemantic-features/index.html)
> [3] [Mechanism for Feature Learning](https://www.science.org/doi/10.1126/science.adi5639)
> [4] [CRAFT: Concept Recursive Activation FacTorization](https://arxiv.org/abs/2211.10154)
> [5] [The Geometry of Truth](https://arxiv.org/abs/2310.06824)
>
> ---
>
> We hope these responses address your concerns. Please let us know if you have additional feedback or comments, and we will be happy to provide further clarification.

---

> > ### Comment · Reviewer_8xsq · 2025-04-03
> >
> > I thank the authors for the detailed rebuttal and additional experiments. This addresses my concern regarding the validation of theoretical findings. I also appreciate the explanation on the relevance to feature learning community. I believe the additional experiments and discussion should be included in the paper to understand the contributions better along with conclusion section discussing limitations and future work.
> >
> > Given the authors's rebuttal, I will increase my score to 4 (accept).
> >
> > Best,
> >
> > Reviewer 8xsq

---

> > > ### Author Response · Authors · 2025-04-03
> > >
> > > We thank the Reviewer for the comments and suggestions and for increasing the score positively. Some of the concerns about relevance and comparison to the literature have led us to rethink the contribution. In the revision of the work, we will include the additional experiments with discussion in the main paper and also write a conclusion section detailing the contribution, future directions, and limitations of the work.
> > >
> > > Thanks,
> > >
> > > Authors

---

### Official Review · Reviewer_tvNN · 2025-03-10

**Overall Recommendation:** 4

**Summary:**

This paper proposes a new problem, inspired by the recent works in ML interpretability, specifically in the literature that posits that activations *linearly* encode concepts. (This is known as the linear representation hypothesis).

The authors suggest that we can use an agent (potentially an LLM) to generate labeled data in representation space. These labels are for triples, and indicate which of two elements of the triple are closest to the third (or equal in the case of a tie).

The goal is essentially to learn a PSD matrix for which the induced distance metric matches all the labeled data.

The authors consider 4 different constraints on the agent's data generating process.

(1) Standard Constructive:  In this case, if the matrix is rank $r$, we can first provide labels that enumerate the nullspace, and then provide the remaining $O(r^2)$ points to learn the low-rank $\Phi$.

(2) Sparse Constructive: The unit vectors and their pariwise sums suffice, but we cannot exploit the rank.

(3) Sampled Activations: Under the assumption each sampled representation can define a corresponding rank-1 matrix, and all are a.s. linearly independent, so the $p(p+1)/2$ can be achieved, to determine $\Phi$

(4) Sparse Sampled Activations: We only get to samples of representations that are at most $s$-sparse, as we would expect in practice. I didn't fully grasp proof--maybe the exposition can be improved.

**Claims And Evidence:**

The main point of this paper is a new theoretical formulation and a solution to that formulation. Given that the proofs in the appendix are correct, (the sketches for at least Thm.1-3 are quite convincing), then there is sufficient evidence of the claims. Experiments also seem to validate the claims.

**Essential References Not Discussed:**

Perhaps some limitations of linear representations could be discussed. See [1]

[1] Recurrent Neural Networks Learn to Store and Generate Sequences using Non-Linear Representations, Róbert Csordás, Christopher Potts, Christopher D Manning, Atticus Geiger

**Experimental Designs Or Analyses:**

Experiments are quite limited. For the final version, the authors might try to work on some additional experiments with greater scale/complexity. However, I don't think the experiments are needed to verify any of the claims, and I recognize that the approaches considered (esp. for Sparse Sampled Activations) might not scale well and require many samples, making them unsuited for real models.

**Methods And Evaluation Criteria:**

Yes. Proofs for formal claims + some small experiments for additional verification.

**Other Comments Or Suggestions:**

I think the "additional experiments" section is interesting, and worthy of inclusion at the end of the paper. I understand the challenge of presenting it though, as it deviated significantly from the approaches discussed in the theory due to practicality.

**Other Strengths And Weaknesses:**

I think the unique perspective of this work is in it's interesting problem framing, using agents to generate labeled data and learning.

An additional weakness is the potential lack of practicality of the first three theorems.

**Questions For Authors:**

1. What is $p_s$? Can you provide some more information in the text?
2. Why is sparse constructive also not $\Theta$ (given the argument)?

**Relation To Broader Scientific Literature:**

This paper has value in its contribution to the theoretical foundation of mechanistic interpretability. The theory and ideas are suggestive of potential new approaches for learning representations, and establish bounds on the complexity of learning from simple comparison data.

**Theoretical Claims:**

I did not check the proofs in the appendix, but the proof sketches for Theorems 1-3 are sufficiently detailed (at least for upper bounds) that I am convinced that the statements are relatively easily proved.

For Theorem 4, I did not follow the proof sketch, and did not check the proof.

---

> ### Author Rebuttal · Authors · 2025-03-30
>
> We thank the reviewer for their thoughtful questions and suggestions. Below, we provide detailed responses to the main concerns.
>
> ---
>
> **1. Potential lack of practicality of the first three theorems.**
>
> As discussed in our response to Reviewer 2 (gr2L), our framework considers general feature learning. The results in Theorems 1–3 apply broadly across domains such as machine teaching, program synthesis, and student–teacher models, where best-case complexity is a central focus. These bounds provide foundational insights and can guide the design of more efficient teaching or interaction systems in these areas.
>
> ---
>
> **2. Discussion of additional experiments in the main text.**
>
> In the revised version of the paper, we will incorporate a substantial discussion of the methodology and findings from the additional experiments directly into the main text.
>
> ---
>
> **3. What is $p_s$? Can you provide more information in the text?**
>
> As detailed in Appendix H, $p_s$ is the probability of the event where $P$ sparse activations (with sparsity $s$) are sampled with a pattern such that the design matrix $\mathbb{M}$ has non-zero determinant (see proof outline of Theorem 4, pp. 7–8). This ensures that at least $P = p(p+1)/2$ linearly independent rank-1 matrices can be constructed from these activations. We will highlight and define this quantity more clearly in the main paper.
>
> ---
>
> **4. Why is sparse constructive also not $\Omega$?**
>
> As noted in the remark following Theorem 2, the lower bound of $\Omega(p^2)$ holds when feedback sparsity is of constant order, i.e., $s = O(1)$. This is sufficient for cases where feedback is restricted to constant-sparse vectors. However, it remains open whether this lower bound remains tight when $s$ grows with $p$, for instance, if $s$ is strongly sublinear in $p$. The upper bound in Theorem 2 is stated for 2-sparse feedbacks, and it applies more generally to higher sparsity values as well.
>
> ---
>
> We hope these responses address your concerns. Please let us know if you have additional feedback or comments, and we will be happy to provide further clarification.

---

### Official Review · Reviewer_gr2L · 2025-03-10

**Overall Recommendation:** 3

**Summary:**

The paper demonstrates that complex features encoded in sparse superposed representations can be effectively learned through a surprisingly minimal and indirect form of feedback. More specifically, the authors show that there is low feedback complexity required to learn sparse superposed features using relative triplet comparisons.

## update after rebuttal
I have carefully reviewed your rebuttal and found that most of my concerns are likely addressable and trust that the authors will be able to improve the draft. I do suggest the authors move the experimental information in Appendix 1 to the main text to demonstrate applicability. I will be increasing my score for the submission.

**Claims And Evidence:**

The authors demonstrate the following differences in complexity when going from a fully general (worst-case) setting to an assumption of low-rank structure:

**Worst-case scenario:**

$$\frac{p(p+1)}{2}$$

- \(p\) denotes the dimension of the representation space

**Improved complexity (low rank):**

$$\frac{r(r+1)}{2} + (p - r) - 1$$

- \(r\) represents the rank of the feature matrix .

Since \(\Phi^*\) is effectively \(r\)-dimensional, one can exploit that low-rank structure to reduce the number of constraints needed to recover \(\Phi^*\) up to a positive scalar factor. Therefore, you only need to describe how the matrix acts on its r-dimensional subspace (where it’s nonzero).

**Essential References Not Discussed:**

I do not have strong familiarity with the theoretical work in this area to know if essential references are missing.

**Experimental Designs Or Analyses:**

Outside of reviewing this paper, the soundness/validity was not checked.

**Methods And Evaluation Criteria:**

The authors take two approaches to evaluate their claims:

**Theoretical Analysis**
They derive upper and lower bounds on the number of feedback constraints required to identify a target feature matrix up to a positive scale factor. By contrasting constructive feedback (where any activation can be chosen) with sampling-based feedback (where activations are drawn from known distributions), the authors show how assumptions like low-rank structure dramatically reduce the needed constraints.

**Empirical Evaluation**
They implement four feedback strategies—Eigendecomposition, Sparse Constructive, Random Sampling, and Sparse Sampling—and test them on synthetic tasks (Recursive Feature Machines) and large-scale sparse autoencoders (e.g., Pythia-70M). Key metrics include mean squared error between the reconstructed matrix and the ground truth, along with the total number of constraints each method requires.

**Other Comments Or Suggestions:**

Table 1: where the row header says “Stardard Constructive” instead of “Standard Constructive.”

**Other Strengths And Weaknesses:**

The reduction in complexity is a desirably property in the space of model interpretability.

Without a discussion section or conclusion, it was very difficulty to properly evaluate this paper. In its current form, the paper feels incomplete. Also the flow of the paper is very difficult to parse given all the different constructions of the derivations.

**Questions For Authors:**

What is the value of the constructive setting outside of theoretical considerations?

How robust are the recovery guarantees when the feedback is noisy or inconsistent?

How sensitive are the results to the assumptions on activation sparsity and the underlying distribution?

**Relation To Broader Scientific Literature:**

The key impact this paper makes appears to be towards the interpretability community. From this perspective, many of the formulations presented in the main text may not apply. For example, the construction setting is not possible to apply to models (like LLMs) since the activation features cannot be controlled in such a manner.

**Theoretical Claims:**

As a reviewer that does not work in the domain of theory, I did not carefully assess the correctness of the proofs in this work.

---

> ### Author Rebuttal · Authors · 2025-03-30
>
> We thank the reviewer for their thoughtful questions and suggestions. Below, we provide detailed responses to the main concerns.
>
> ---
>
> **1. Evaluation of the paper.**
>
> We emphasize the broad relevance of our theoretical work
>
>  -  They apply to feature extraction in LLMs (see Section 6 and Appendix I for additional experiments for LLMs).
>  -  Our results bound sample/feedback complexity for any class of dictionaries, e.g., sparse ones.
>  -  Furthermore, they inform directions for mechanistic interpretability based on Neural Feature Ansatz. [1], [2]
>
>
> **2. Lack of a discussion section or conclusion.**
>
> In the revision, we will include a dedicated discussion section that highlights the main contributions, draws connections to related approaches in feature learning, and outlines the limitations of our work.
>
> ---
>
> **3. What is the value of the constructive setting outside of theoretical considerations?**
>
> Our framework encompasses general feature learning and applies to both standard and sparse samples/activations. The constructive setting can be useful in several practical contexts:
>
> - **Machine teaching and program synthesis:**
>   There is growing interest in understanding the *teachability* of a hypothesis space. Analyzing the best-case complexity of teaching is essential in settings where an agent selects samples to guide a learner. Similarly, in program synthesis, constructive feedback or targeted examples can be valuable for efficiently steering a learner to a fixed goal. [3], [4]
>
> - **Student-teacher models:**
>   The bounds in the constructive setting translate directly into guarantees for interactive learning in student-teacher models. While the setting may appear contrived, the resulting bounds clarify the gap in informativeness between different types of feedback. [5]
>
> ---
>
> **4. How robust are the recovery guarantees when the feedback is noisy or inconsistent?**
>
> This is an interesting direction for future work. There are two settings worth exploring:
>
> - **Weak teaching agent:**
>   Here, the agent provides feedback with some noise (e.g., Gaussian). This leads to an additional error-dependent factor in the feedback complexity, on average. Note that the reduction in Lemma 2 (p. 4) assumes that feedback vectors are equivalent in norm under the feature map, so exact recovery is not possible in the noisy setting. The increase in feedback complexity is due to difficulties in reliably sampling low-error feedbacks, which could be addressed via concentration bounds under specific noise models.
>
> - **Learner receives noisy feedback:**
>   When the learner only receives relative (and noisy) feedback on activations, standard information-theoretic arguments imply that, in the worst case, the feedback complexity is unbounded. Specifically, the differences between rank-1 matrices generated by noisy feedbacks must lie in the orthogonal complement of the target feature matrix in the space of symmetric matrices—an event that occurs with zero probability in general (see related argument in Section 5). It would be interesting to study whether this can be improved in interactive settings or under structured assumptions on the feature matrix.
>
> ---
>
> **5. How sensitive are the results to the assumptions on activation sparsity and the underlying distribution?**
>
> We analyze two sampling setups:
>
> - **Random sampling:**
>   We assume the input distribution follows a general Lebesgue measure, and we apply Sard's Theorem to derive tight bounds. Thus, unless one considers more exotic measures, the result is robust across a wide class of distributions. Since Lebesgue measures cover most distributions of practical and theoretical interest, the bounds provide a meaningful characterization of feedback informativeness in this setting.
>
> - **Sparse sampling:**
>   For sparse activations, we follow the distributional assumptions common in the dictionary learning literature (cf. Gribonval et al., 2015). The upper bound in Theorem 4 (p. 7) does not depend on a specific distribution. Instead, it uses a pattern-matching argument and Hoeffding's inequality. The bound depends on the sparsity level and the probability of non-zero coordinates in the activation vectors. Thus, feedback complexity is lower for less sparse activations and for distributions with higher probability mass on more dimensions.
>
> ---
>
> ### **References**
>
> [1] [Mechanism for Feature Learning](https://www.science.org/doi/10.1126/science.adi5639)
>
> [2] [Aggregate and conquer](https://arxiv.org/pdf/2502.03708)
>
> [3] [An Overview of Machine Teaching](https://arxiv.org/pdf/1801.05927)
>
> [4] [Program Synthesis](https://www.microsoft.com/en-us/research/wp-content/uploads/2017/10/program_synthesis_now.pdf)
>
> [5] [Teacher-Student Architecture for Knowledge Distillation: A Survey](https://arxiv.org/abs/2308.04268)
>
>
> ---
>
> We hope these responses address your concerns. Please let us know if you have additional feedback or comments, and we will be happy to provide further clarification.

---

> > ### Comment · Reviewer_gr2L · 2025-04-07
> >
> > Thank you for your responses. I have carefully reviewed your rebuttal and found that most of my concerns are likely addressable and trust that the authors will be able to improve the draft. I do suggest the authors move the experimental information in Appendix 1 to the main text to demonstrate applicability. I will be increasing my score for the submission.

---

> > > ### Author Response · Authors · 2025-04-07
> > >
> > > Dear Reviewer,
> > >
> > > Thanks for increasing the score positively. We agree to include a dedicated discussion section and experimental information in the main text.
> > >
> > > Thanks,
> > >
> > > Authors

---

### Official Review · Reviewer_wsQ7 · 2025-03-23

**Overall Recommendation:** 3

**Summary:**

This paper investigates theoretical bounds on feedback complexity for learning feature matrices through triplet comparisons. The authors analyze both constructive settings (where agents select activations) and distributional settings (with sampled activations). In particular, for a rank-r feature matrix in p-dimensional space, they prove bounds of Θ((r(r+1)/2) + (p-r)) comparisons in the constructive setting and Θ(p(p+1)/2) for general activations sampled from a Lebesgue distribution (alongside bounds for sparse sampling settings). They validate their findings using Recursive Feature Machine-trained models and dictionaries from sparse autoencoders trained on language models like Pythia-70M and Board Game models.

**Claims And Evidence:**

The paper's theoretical claims are supported by mathematical development in the main text, with detailed proofs referenced in appendices.
The authors establish a learning framework using triplet comparisons, which they reduce to pairwise comparisons with equality constraints in Lemma 2. This allows them to reformulate the learning problem in terms of matrices that annihilate a specific subspace, providing a geometric interpretation.

For constructive feedback settings, Theorem 1 proves tight bounds of Θ((r(r+1)/2) + (p-r) ) for rank-r matrices through a decomposition approach. For general sampling, Theorem 3 establishes bounds of Θ(p(p+1)/2) using Lebesgue measure properties. For sparse sampling, Theorem 4 provides an upper bound based on probability theory.

The experimental validation shows alignment with theoretical predictions. Figure 1 demonstrates that for a 10×10 feature matrix, eigendecomposition-based and sparse constructive methods achieve the predicted performance. Figure 2 shows how sparsity probability affects feedback requirements, with higher sparsity requiring more samples as predicted, with some additional results on language models like Pythia-70M and Board Game models in Appendix.

**Essential References Not Discussed:**

Some relevant recent works that could strengthen the paper:
1. Lan et al. (2024) "Sparse Autoencoders Reveal Universal Feature Spaces Across Large Language Models" (arXiv:2410.06981), which shows how SAEs transform LLM activations into interpretable spaces where feature universality can be studied.
2. Radhakrishnan et al. (2024) "Linear Recursive Feature Machines provably recover low-rank matrices" (arXiv:2401.04553), which provides theoretical guarantees for RFMs in low-rank matrix recovery.
3. Recent work from Anthropic on scaling sparse autoencoders to larger models and using them for causal interventions, that demonstrate practical applications for dictionary learning.

**Experimental Designs Or Analyses:**

The experimental design includes both synthetic and language model-based evaluations:
1. RFM Experiments: The authors use monomial regression with RFMs to generate controlled test cases. They compare four feedback methods (eigendecomposition, sparse constructive, random sampling, sparse sampling) on 10×10 matrices of rank 4.
2. Sparsity Analysis: Figure 2 investigates how sparsity probability affects feedback requirements, showing increased requirements with higher sparsity.
3. Language Model Dictionaries: The paper mentions experiments on dictionaries from Pythia-70M and Board Game models with dimensions of 32k×512 and 4096×512 respectively, using batch-wise gradient descent for optimization.

The authors are encouraged to include experiments and discussions based on LLM dictionary learning in the main paper to demonstrate how the ideas work in realistic settings and at scale.

**Methods And Evaluation Criteria:**

The paper establishes a well-defined mathematical framework for analyzing feedback complexity. The authors use an oblivious learner model (Definition 3) that randomly selects a feature matrix satisfying all feedback constraints, allowing them to focus on information-theoretic aspects. For evaluation, they use mean squared error (MSE) between learned and ground truth feature matrices to assess reconstruction quality. The experiments employ Recursive Feature Machines (RFMs) to generate controlled test cases with known ground truth matrices. The monomial regression task f*(z) = z₀z₁1(z₅ > 0) provides a controlled environment for testing. For large-scale experiments, the authors mention using batch-wise gradient descent for optimization with dictionaries from language models.

**Other Comments Or Suggestions:**

- More detailed analysis of language model experiments would strengthen the paper.
- Discussion of adaptive feedback mechanisms could enhance future directions.
- More explicit connection to interpretability applications would increase impact.

**Other Strengths And Weaknesses:**

N/A

**Questions For Authors:**

1. For the sparse sampling scenario, do you have insights on potential matching lower bounds to complement your upper bound? Not sure if I missed some details here.
2. In your language model dictionary experiments, how well did the empirical feedback requirements align with your theoretical predictions? What specific challenges did you encounter at this scale?
3. How might your theoretical framework extend to adaptive feedback scenarios, where triplets are selected based on previous responses? Could this potentially reduce the number of comparisons needed in practice?

**Relation To Broader Scientific Literature:**

The paper connects to several research domains:

1. Dictionary learning: The work extends dictionary recovery approaches by focusing on learning from feedback rather than direct samples, building on prior work by Gribonval & Schnass (2010) and Arora et al. (2013).

2. Mechanistic interpretability: The paper contributes to understanding feature representation in neural networks, relating to work on sparse autoencoders for interpretability by Bricken et al. (2023).

3. Superposition theory: The work addresses neural network superposition as described by Elhage et al. (2022), where models represent more features than dimensions through sparse linear combinations.

4. Mahalanobis distance learning: The triplet comparison framework connects to work on metric learning by Kulis (2013) and Schultz & Joachims (2003).

5. Recursive Feature Machines: The experiments leverage recent work on RFMs by Radhakrishnan et al. (2024), which use the Average Gradient Outer Product for feature learning.

**Theoretical Claims:**

The paper presents several theoretical results that I reviewed at a high level, though I did not work through all the mathematical details:

1. The key theoretical contribution is establishing bounds on the minimum number of triplet comparisons needed to learn feature matrices under different settings. The authors develop a framework where triplet comparisons are reduced to pairwise equality constraints, then analyze the problem through the lens of orthogonal complements in matrix spaces.
2. Their approach decomposes feature matrices into eigenspace and null space components, allowing them to derive tight bounds for the constructive setting. For distributional settings, they leverage measure theory to establish bounds for general activations and probability theory for sparse activations.
3. These theoretical developments build upon established results in linear algebra, measure theory, and probability, though a complete verification would require detailed examination of the appendices.

---

> ### Author Rebuttal · Authors · 2025-03-29
>
> We thank the reviewer for their thoughtful questions and suggestions. Below, we provide detailed responses to the main concerns.
>
> ---
>
> **1. Potential lower bound to complement the sparse sampling upper bound (Theorem 4)**
>
> We conjecture that the dependence on $p^2$ and $\log \tfrac{1}{\delta}$ in the feedback complexity bound of Theorem 4 is fundamental. A matching lower bound is likely to follow from standard statistical lower bound techniques, such as Fano’s inequality. Specifically, identifying a feature matrix from sparse feedback observations constitutes a high-dimensional hypothesis testing problem, where each feedback vector reveals limited information. Under such settings, Fano-style arguments imply that at least $\Omega(p^2 \log \tfrac{1}{\delta})$ feedbacks are necessary to achieve reliable identification with confidence $1 - \delta$.
>
> Additionally, the quantity $p_s$ in Theorem 4, which captures the number of monomial components affected by sparse activations, could potentially be refined. A sharper bound may be achievable by analyzing high-probability events where sampled sparse activations span the full $p^2$-dimensional space of symmetric matrices.
>
> We note that deriving a fully rigorous tight lower bound is valuable future work.
>
> ---
>
> **2. Empirical vs. theoretical feedback complexity in large-scale SAE experiments**
>
> In Table 2 (Additional Experiments, p. 29), we report the empirical feedback complexity and the corresponding Pearson Correlation Coefficient (PCC) for various feedback mechanisms on the ChessGPT SAE. For Eigendecomposition, Sparse Constructive, and Random Sampling, the empirical observations align closely with the theoretical bounds stated in Theorems 1–3. For Sparse Sampling, we used 3-sparse activations, which result in a small $p_s$. Consequently, the PCC improves with increasing feedback but remains lower than the other mechanisms due to the fixed sparsity relative to the input dimension $p = 4096$. We observe similar patterns on Pythia-70M SAEs.
>
> To further support the comparison between theoretical and empirical bounds, we include a synthetic experiment designed to highlight the alignment between the two (see response in bullet point 1. to Reviewer 8xsq).
>
> ---
>
> **3. Challenges in scaling to large dimensions**
>
> For models such as ChessGPT and Pythia-70M, the total number of degrees of freedom $P = p(p+1)/2$ reaches 8.3 million and 512 million, respectively. These scales present significant challenges in terms of storage and computation. Except for Eigendecomposition—which exploits the low-rank structure of the feature matrix—all feedback mechanisms require complexity scaling with $P$.
>
> To address this, we implemented a sparse feedback representation and avoided storing full dense vectors. Without constant sparsity, storing even $10^7$ feedback vectors becomes impractical. Furthermore, solving the resulting linear systems via direct constraint satisfaction is infeasible. Instead, we use a batch-wise gradient descent method (Algorithm 3, p. 29) to solve for the feature matrix using the feedback generated by rank-1 outer products.
>
> ---
>
> **4. Extending to adaptive feedback settings**
>
> We agree that extending our framework to incorporate adaptive feedback is a compelling direction for future work. In our current setting, the learner is *oblivious*—it selects a feature matrix uniformly at random from the set of hypotheses consistent with received feedback. As a result, the teacher cannot leverage adaptivity across feedbacks.
>
> However, in more expressive settings where the learner uses a *preference function* to select hypotheses (e.g., minimizing Frobenius norm, promoting sparsity, or favoring identity-like solutions), adaptive feedback strategies could substantially reduce feedback complexity. Related ideas have been explored in recent work on machine teaching and interactive learning:
>
> - [Teaching via Best-Case Counterexamples in the Learning-with-Equivalence-Queries Paradigm](https://openreview.net/forum?id=Ee7IOrpLwT)
>
> - [Preference-Based Batch and Sequential Teaching: Towards a Unified View of Models](https://papers.nips.cc/paper_files/paper/2019/hash/4dc3ed26a29c9c3df3ec373524377a5b-Abstract.html)
>
> Incorporating such learner models into our framework would allow for interaction-aware teaching agents and could yield tighter feedback complexity bounds. This remains an exciting avenue for future investigation.
>
> ---
>
> **5. Essential References**
>
> We thank the reviewer for highlighting these relevant references. We will incorporate them into the appropriate sections of the revised paper. Additionally, we note that the work of Radhakrishnan et al. (2024) addresses low-rank matrix recovery under exact measurement conditions, whereas our setting involves relative feedback, which introduces additional complexity.
>
> ---
>
> We hope these responses address your concerns. Please let us know if you have additional feedback or comments, and we will be happy to provide further clarification.

---

### Decision · Program_Chairs · 2025-05-01

**Decision:**

Accept (poster)

**Comment:**

This paper received positive feedback from all reviewers, in particular based on the originality of the proposed approach and the clear theoretical results. The main criticisms are that additional experiments from the appendix should be added to the main paper, and adding a discussion section would help understand the impact of this work. Given that the authors plan to fix these two points, I recommend accepting this paper.

Additionaly, I think that the exposition of the problem could be further improved, in particular by paragraph l.69, making it clear which part is fixed for the model (the dictionary $D$ from my understanding) and which part is learn with the feedback ($\hat D$, which is then also denoted $D$ and changes to $L$ later on). The overall flow of the paper can also be improved (**gr2L**) by stating more clearly the selection mechanisms considered in l.74, and giving a better description of how the label $l$ is constructed.